# A geodatabase of historical landslide events occurred in the highly urbanized volcanic area of Campi Flegrei, Italy

Giuseppe Esposito[1], Fabio Matano[2]

[1]National Research Council - Research Institute for Geo-hydrological Protection (CNR-IRPI), Perugia, 06128, Italy
[2]National Research Council - Research Institute of Marine Sciences (CNR-ISMAR), Napoli, 80133, Italy

*Correspondence to*: Giuseppe Esposito (giuseppe.esposito@irpi.cnr.it)

**Abstract.** The analysis of geological processes threatening people and properties in a populated region should be based on a comprehensive knowledge of historical events and related characteristics. This type of information is essential for predisposing event scenarios, validating prediction models, and planning risk mitigation measures. Such activities may be more complex in
some geological settings, where urban settlements are exposed to multi-hazard conditions. This is the case of the densely populated Campi Flegrei volcanic area located in the Campania region, southern Italy. Here, volcanic and seismic hazards are associated with landslides, floods, and coastal erosion, which are playing a relevant role in the landscape modification. The CAmpi Flegrei LAndslide Geodatabase (CAFLAG), here presented, provides information related to 2302 landslides that occurred in the continental, coastal and insular sectors of the study area, during the 1828-2017 time interval. Data associated
to the collected landslide events highlight the characteristics of both landslides and of the affected sites. Most of the cataloged mass movements consist of rock falls affecting rocky slopes formed by lithified volcanic rocks, such as tuff or ignimbrite. In addition, rainfall-induced earth and debris slides translating into rapid flows or debris avalanches are widespread along steep slopes mantled by weakly welded pyroclastic airfall deposits, similarly to other areas of the region. The highest density of landslides concentrates along the coastline where mass movements are contributing to the retreat of coastal cliffs, and along
inland slopes exposed towards the western directions from which most of the storm systems come from. Temporal information shows peaks of landslides in the years 1986, 1997, 2005. A total of 127 people lost their life as a consequence of 53 cataloged landslides, with a frequency of deadly events observed however to decrease since the early 1980s. This information will be useful to analyze mortality and risk conditions still affecting population of the Campi Flegrei caldera, which require to be fully addressed with advanced knowledge and accurate scenarios. The full database is freely available online at
https://doi.org/10.4121/14440757.v2 (Esposito and Matano, 2021).

# 1 Introduction

Landslides are among the most effective agents in the landform evolution, especially in areas affected by active volcano-tectonic processes. In these areas, besides volcanic and associated seismic risks, hillslope instability processes pose an additional risk to the exposed urban settlements, also during non-active volcanic phases. Landslides contribute to the dismantling of volcanic edifices by displacing rock masses that form the volcanic flanks (Siebert and Roverato, 2021; Di Traglia et al., 2020; Williams et al., 2019; Walter et al., 2019; Oehler et al., 2004; Ablay and Hürlimann, 2000), involving in some cases the submarine domain (Dufresne et al., 2021; Coombs et al., 2007; Masson et al., 2006; Watts et al., 2012; Casalbore et al., 2020; Chiocci et al., 2008). Weakly welded pyroclastic deposits covering volcanic or non-volcanic slopes can be also mobilized in response to rainfall or snow and ice melting by means of rapid or extremely rapid debris and hyperconcentrated flows, occurring both simultaneously with volcanic activity (syn-eruptive lahars) and during pauses or volcano dormancy, i.e., "post-eruptive lahars" (Lavigne and Thouret, 2002; Capra et al., 2004; Pierson et al., 2014; Thouret et al., 2020). In addition, similar instability conditions can affect wildfire-affected hillslopes covered by volcanic soils (Neris et al., 2013; Esposito et al., 2019; Peduto et al., 2022).

In Italy, the densely urbanized area of Campi Flegrei (Fig. 1) corresponds to an active volcanic caldera considered among those with the highest volcanic risk in the world (De Natale et al., 2006). For this reason, scientific research has been mostly focused on volcanic and seismic hazards, making the Campi Flegrei one of the most monitored and analyzed caldera in the world, and a reference to enhance the knowledge on mechanisms controlling volcanic unrest (Troise et al., 2019). On the other hand, relatively poor attention has been paid towards exogenous processes such as landslides, floods, and coastal erosion. As a result, the risk posed by landslides in the Campi Flegrei volcanic area is currently underestimated both among the scientific community and the population. Landslides in this area have been reported since the Imperial Roman period (Di Martire et al., 2012). Among the most disastrous recent events documented in the literature, it deserves to mention: 1) the huge tuff collapse occurred along Mt. Echia (Naples) in 1868, causing 60 victims and dozens of injuries (Calcaterra and de Luca Tupputi Schinosa, 2006); 2) flash floods, debris flows, and rock falls occurred in Casamicciola, Lacco Ameno and Forio (Ischia island) on 24 October 1910, causing 6 fatalities and widespread damage (Santo et al. 2012); 3) the cliff failure occurred along the Maronti beach in 1978, killing five tourists (Del Prete and Mele, 1999); 4) a series of flow-like mass movements that in 2006 hit some buildings at the footslope of the Monte Vezzi (Ischia island), where four people died (De Vita et al., 2007; Di Nocera et al., 2007). These and other minor deadly events indicate that landslides pose a serious societal risk in the Campi Flegrei area (Cascini et al., 2008). This risk has been also confirmed by the recent landslides of 26 November 2022 at Casamicciola, in the Ischia island, where 12 people lost their life (https://polaris.irpi.cnr.it/event/colate-di-fango-e-detrito-a-casamicciola-terme-isola-di-ischia/). Such a further event has highlighted as Ischia can be considered a kind of hotspot for the geo-hydrological risk in the area.

The compilation of an up-to-date and complete landslide database that records location, types and, where known, the date of occurrence of mass movements is the simplest initial approach to any study of landslide hazard (Carrara et al. 1995; Soeters

and van Westen 1996; Devoli et al., 2007; Guzzetti et al. 2012; Ardizzone et al., 2023). As underlined by Napolitano et al. (2018), information on historical landslides is important to understand the complexities and dynamics of past events, as well as to construct and validate landslide prediction models able to support the designing of appropriate mitigation measures. In the Campi Flegrei area, preliminary landslide inventories have been provided by Beneduce et al. (1988) and Calcaterra et al. (2003a). Dozens of events have been also encompassed in databases and inventories realized at national scale, based on archive

research (AVI database; Guzzetti et al., 1994) and aerial photo interpretation at 1:25.000 scale (IFFI landslide inventory map; Trigila et al., 2010). With reference to the city of Naples, the landslide activity has been reconstructed by Calcaterra et al. (2002, 2003a, 2007) and Di Martire et al. (2012) by means of archival and bibliographic research, as well as by Miele et al. (2022). Failures and retreat rates affecting the cliffed coastline have been investigated by Matano et al. (2016), Esposito et al. (2017, 2018a, 2020), and by Caputo et al. (2018). Specifically, decadal retreat rates have been quantified in the order of 1.20

m/yr (i.e. Torrefumo cliff during 1956-74), and short-term (annual) retreat rates within 0.01-0.10 m/yr on average (i.e. Coroglio cliff during 2013-2015, and Torrefumo cliff during 2013-2016).

With the aim of providing a comprehensive landslide geodatabase of the Campi Flegrei area referred to the 1828-2017 time interval, we have gathered all the landslide-related information made available from several sources, including a series of events collected by means of local press, news websites and fieldworks. Data have been organized in a GIS environment to

guarantee an easy access, management, updating and sharing with different users. All the collected and revised data have been used to develop a series of statistics about spatial and temporal distribution of the events, failure types, impact, and relationships with geological and geomorphological properties of the affected hillslopes. The CAmpi Flegrei LAndslide Geodatabase (CAFLAG) here described may be used for future analyses aimed at evaluating the landslide susceptibility, hazard, and risk conditions, as well as to understand the geomorphic evolution of the study area. In addition, the CAFLAG data may be of

relevant interest for the international landslide research community to understand landslide dynamics in volcanic settings during dormant phases, and to implement specific numerical models aimed at evaluating the landslide susceptibility in similar areas.

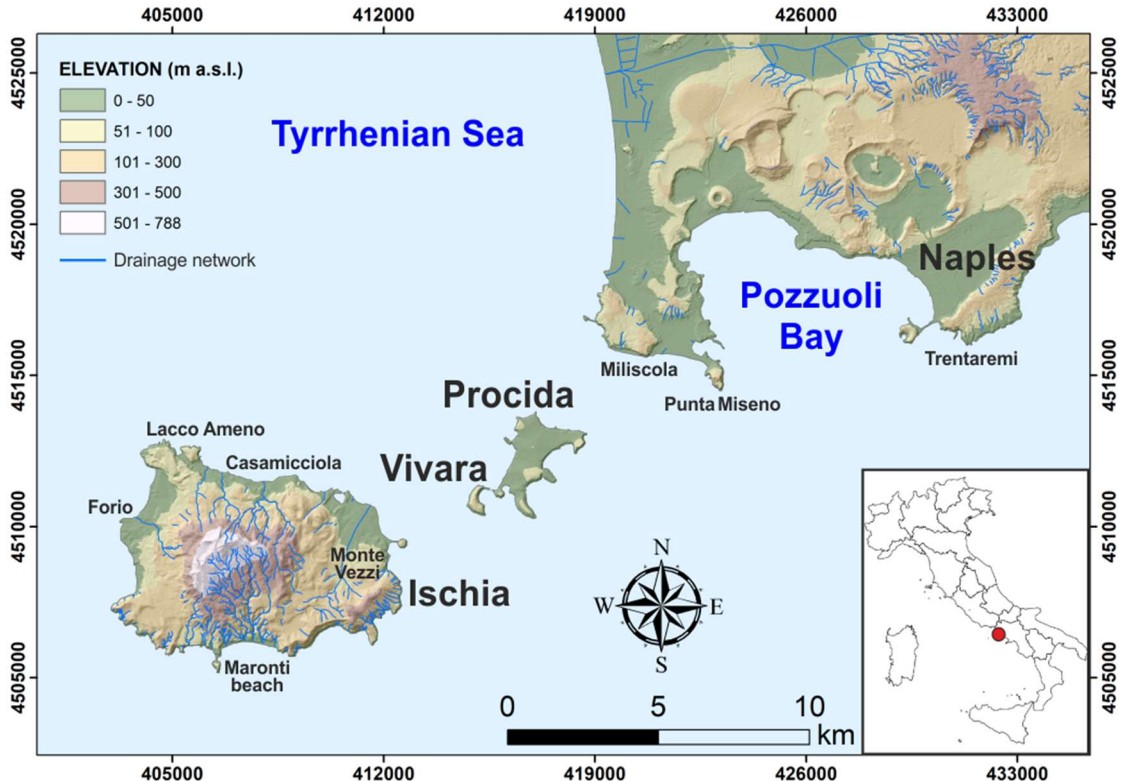

**Figure 1: Location (red dot in the inset map) and shaded relief of the Campi Flegrei volcanic area including the islands of Ischia, Procida, and Vivara (coordinates in EPSG 32633). The shaded relief has been developed from elevation data of the *SIT Regione Campania (for reference see the main text).***

## 2 Study area

The Campi Flegrei (Fig. 1) is an active volcanic area located in southern Italy, within the central part of the Campanian Plain, a graben-like structure resulted from the tectonic displacement of a Mesozoic carbonate basement. The Campi Flegrei corresponds to a quasi-circular depression extending for about 200 km$^2$, a large part of which develops off the Pozzuoli Bay (Sacchi et al., 2011, 2020). The insular sector is represented by the islands of Ischia, Procida and Vivara (Fig. 1).

The most ancient volcanic activity is associated to the volcanism of Ischia, probably started more than 150 ka BP (Poli et al., 1987; Vezzoli, 1988), whereas the activity of the Procida island occurred mostly 70 ka BP (De Astis et al., 2004). The emplacement of the Mt. Epomeo Green Tuff Formation (about 55 ka BP; Vezzoli, 1988) formed the backbone of the highest relief in the area (actually 788 m a.s.l. in the Ischia island). About 39 ka BP, the catastrophic eruption of the Campanian Ignimbrite took place, causing the formation of a large caldera (Rosi and Sbrana, 1987; Orsi et al., 1996; Perrotta et al., 2006) that was reshaped by the more recent phreato-plinian eruption of the Neapolitan Yellow Tuff (NYT), dated at 15 ka BP (Scarpati et al., 1993; Insinga et al., 2004; Deino et al., 2004). The volcanic activity that occurred after the NYT eruption was subdivided in three main epochs (Di Vito et al., 1999), which were interrupted by long periods of quiescence. In these epochs,

minor explosive events occurred within the rim of the NYT caldera, creating at least 52 monogenic phreato-magmatic vents, including tuff rings, tuff cones, cinder and spatter cones (Di Vito et al., 1999; Insinga et al., 2006; Perrotta et al., 2011). The latest eruption, known as "Monte Nuovo Eruption", occurred in the 1538 A.D. Currently, the Campi Flegrei is characterized by very high levels of volcanic risk due to many towns lying in a caldera characterized by unrest conditions, as demonstrated by widespread fumaroles, thermal springs, earthquakes and ongoing ground deformation.

The current geomorphological configuration of the continental and insular sectors of the Campi Flegrei is the result of volcanic and geomorphic processes occurred in the last 15 ka, after the NYT eruption. Relics of volcanic edifices partially dismantled by sea erosion and landslides can be found along or near the present coastline and in the submarine part of the NYT caldera (e.g. Sacchi et al., 2009; 2011). Landslides due to earthquakes, rainstorms, marine erosion or human actions have repeatedly affected the slopes of the inland volcanos, ravines and streams, as well as the steep scarps of gullies and coastal cliffs through time (Ducci & Napolitano, 1994; Guadagno and Mele, 1995; Mele and Del Prete, 1998; Del Prete and Mele, 1999, 2006; De Vita et al., 2006; Santo et al., 2012; Esposito et al., 2018a, 2020). Besides volcanic morphologies, the geomorphic setting of Campi Flegrei also includes lowlands that are located between volcanic edifices or close to the NYT caldera rims.

The drainage network is mostly arranged in radial patterns typical of volcanic edifices, with isolated short channels along gentle slopes and plains. Vegetation covering the volcanic hillslopes consists mainly of Mediterranean shrubs that alternate with vineyards and cultivated areas, and locally with pine woods. In addition to climatic conditions described in the following section, the widespread vegetation is fostered also by fertile soils (andosols) developed from pyroclastic deposits.

Currently, the study area is inhabited by approximately 500,000 people, with an average population density of 3440 people/km$^2$. Besides, it is visited each year by thousands of tourists due to the archaeological, cultural, and environmental heritages.

## 2.1. Weather and climate

The study area is characterized by a Mediterranean climate with hot, dry summers and moderately cool rainy winters. Mean annual temperatures are in the range of 10° at the hilly altitudes, 18°C along the coastline, and 15.5°C in the plains surrounding the inland reliefs (Ducci and Tranfaglia, 2005). The mean temperature of the warmest month (July) ranges between 24-28°C, whereas the mean temperature of the coldest month (January) ranges between 4°-6°C. The rainfall regime is characterized by the maximum amounts in autumn/winter, with a mean annual precipitation of about 700 mm. It is worth noting that during the summer and autumn seasons, the coastal sector is often hit by convective cells forming offshore. Such cells are able to release high amounts of rain in short times, with maximum 10 minutes rain rates higher than 100 mm/h, as highlighted by Esposito et al. (2015) and Fortelli et al. (2019). Waterspouts in front of the Flegrean coast are also associated with these cells. Winds blow mainly from three directions: West, North-East, South. They can reach wind gusts up to 140 km/h in the winter season, able to generate sea storms leading to severe damage on coastal settlements (Fortelli et al., 2021).

## 3 Data and methods

The CAFLAG geodatabase refers to the Campi Flegrei area (Fig. 1), extending about 230 km$^2$ and including the western part

of the city of Naples and the islands of Ischia, Procida, and Vivara. An overview of the attributes associated to each cataloged landslide event is shown in Fig. 2. Information related to both landslide (i.e., data source, localization, date of occurrence, type of mechanism, volume of displaced material, impact on people and properties, predisposing and triggering factors) and the corresponding affected site (i.e., geomorphological and engineering geological properties, landslide susceptibility and hazard levels) is provided where available.

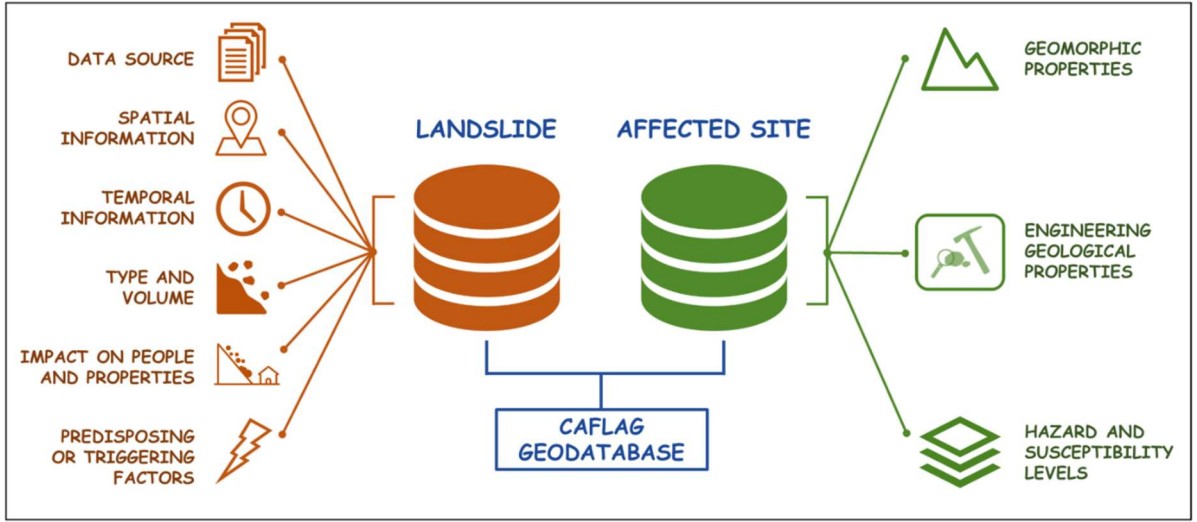


**Figure 2: Overview of the information provided within the CAFLAG geodatabase.**

### 3.1 Landslide data sources

All the considered landslide data sources are listed as follows. The first seven sources can be considered as "archive" sources, whereas the last two as "new" sources:


- IFFI landslide inventory map (*Inventario Fenomeni Franosi in Italia*) (Trigila et al., 2010) (www.progettoiffi.isprambiente.it/en/ accessed on 26 May 2022);

- AVI Catalog (*Aree storicamente Vulnerate in Italia da calamità geologiche ed idrauliche*) (sici.irpi.cnr.it/avi.htm, accessed on 26 May 2022);

- Landslide database and inventory map of the hydrographic Basin Authority (AdB) (namely Campania Centrale Basin Authority) (www.distrettoappenninomeridionale.it/index.php/elaborati-di-piano-menu/bacini-reg-nord-occidentali-bacino-reg-sarno-ex-adb-reg-campania-centrale-menu/piano-assetto-idrogeologico-rischio-da-frana-menu, accessed on 26 May 2022);

- Geological map of Italy - CARG (*Carta Geologica d'Italia*) project (Sheets 446-447, 464, 465; 1:25.000-1:50.000 scale) (www.isprambiente.gov.it/Media/carg/campania.html, accessed on 26 May 2022);
- Landslide inventories of the city of Naples (Calcaterra et al. 2002, 2003a, 2007; Di Martire et al., 2012);
- Seismically induced landslide inventory of the Ischia island (Caccavale et al., 2017, Rapolla et al., 2012);
- Scientific papers (Beneduce et al., 1988; Calcaterra et al., 2003b, 2010; Di Nocera et al., 2007; Santo et al., 2012);
- Field geomorphological surveys performed on coastal sectors during 2013-2018, documented by Esposito et al. (2015, 2017, 2018a, 2020) and Caputo et al. (2018);
- Landslide reports collected from digital archives of the principal newspapers and news websites in the study area (https://napoli.repubblica.it/; www.ilmattino.it; www.cronacaflegrea.it; www.montediprocida.com; www.freebacoli.net; accessed several times during 2013-2018).

All of the events contained in the CAFLAG geodatabase are geocoded in a point shapefile, with each point referring to the centroid of the landslide scar. Pre-existing catalogues (IFFI, AVI, AdB) were available in shapefile formats, so that each event was already associated to a georeferenced point. The other sources, instead, provided static map images requiring a manual digitalization of the location points, or information allowing to place the representative point on the corresponding hillslope, by using as support the WMS services of the Campania Region webgis (https://sit2.regione.campania.it/content/servizi-wms, accessed on 26 May 2022), such as: topographic map at 1:5000 scale; ii) the derived Digital Terrain Model (DTM) with a pixel size of 5 m; iii) and orthophoto at 1:10000 scale. All these datasets refer to the years 2004-2005.

Given that the used sources included sometimes redundant events, a preliminary data filtering was performed manually by deleting those characterized by the same attributes and location, indicating therefore the same landslide event. Generally, the level of completeness of information associated to the collected events is quite heterogeneous, as well as the level of accuracy, as explained in the following section.

## 3.2 Geodatabase structure

Attributes associated to the landslide events of the CAFLAG geodatabase (Fig. 2) are specified in Table 1. The primary set of information refers to data sources, spatial and temporal properties of the events with related accuracy. High spatial accuracy was attributed to events associated with accurate geographical information, as for example accurate spatial coordinates provided in available databases, indications about the affected road section, specific buildings or sites. Where this type of information was not available, a low spatial accuracy was indicated, providing rough coordinates of the landslide-affected site (e.g. center of the locality, road or beach hit by the mass movement). High temporal accuracy was attributed to events associated with a complete date (i.e. dd/mm/yyyy), moderate accuracy to events characterized by both month and year of occurrence, and low accuracy to events which could be placed in a wide period of time, before or within a specific year. The movement type (fall, flow, slide, complex) and the involved material (earth, debris, rock) were defined according to the

classification of Cruden and Varnes (1996). Further attributes describe the impact of the cataloged landslides on people, buildings and infrastructures, and in a few cases only, indications about the factors predisposing or triggering the mass movements (e.g., rainfall, digging activities).

A secondary set of information refers to geomorphological properties of the landslide-affected sites, such as: i) type and name
of the affected watersheds; ii) geomorphological context in terms of inner slope (i.e. not related to direct sea action), coastal slope, or quarry; iii) and morphometric properties of the affected hillslopes (elevation, slope, aspect), calculated from the intersection of each vector point with the DTM. By overlapping the point shapefile with engineering geological maps available for the area (Caccavale et al., 2017; Sacchi et al., 2015), both geotechnical and seismic properties of lithotypes involved in the landslides were obtained, as well as the landslide hazard and susceptibility levels of the sites, extracted from the Landslide
Risk Plan of the Campania Centrale Basin Authority (www.distrettoappenninomeridionale.it/index.php/elaborati-di-piano-menu/bacini-reg-nord-occidentali-bacino-reg-sarno-ex-adb-reg-campania-centrale-menu/piano-assetto-idrogeologico-rischio-da-frana-menu, accessed on 26 May 2022).

All the statistical analyses were performed by means of the Microsoft$^{TM}$ Excel software, and results were displayed in a series of histograms and pie-charts. Statistics related to deadly events, based on the Frequency-Number relations, were aimed at
analyzing the societal risk by following the work of Cascini et al. (2008).

**Table 1 – Attribute list of the CAFLAG geodatabase.**

| ATTRIBUTE | BRIEF DESCRIPTION | TYPE | FEATURES (range, classes, measure unit, etc.) |
|---|---|---|---|
| ID | A unique, incremental number as identifier for each landslide event | numeric | 4 digits |
| SOURCE_T | Typology of data source | text | 2 classes (archive, new) |
| SOURCE | Name of data source | text | free |
| LATITUDE | Latitude of the landslide-related vector point converted to UTM coordinate Y | UTM km coordinate | EPSG:32633 |
| LONGITUDE | Longitude of the landslide-related vector point converted to UTM coordinate X | UTM km coordinate | EPSG:32633 |
| TOWN | Affected town | text | 13 towns |
| LOCALITY | Affected locality (local name of the involved area) | text | free |
| LOCAT_ACC | Location accuracy | text | 2 classes (low, high) |
| TEMP_IND | Temporal indication | text | 4 classes (time period, year, partial or full date of occurrence) |
| TEMP_ACC | Temporal accuracy | text | 3 classes (low, moderate, high) |
| YEAR | Year of occurrence | numeric | 4 digits (yyyy) |
| MONTH | Month of occurrence | numeric | 2 digits (mm) |
| DAY | Day of occurrence | numeric | 2 digits (dd) |

| TIME | Time of occurrence | Time or day phase | hh:mm or 3 classes (morning, afternoon, night) |
|---|---|---|---|
| MOV_TYPE | Type of movement according to the classification of Cruden and Varnes (1996) | text | 4 classes (fall, flow, slide, complex) |
| MAT_TYPE | Type of displaced material according to the classification of Cruden and Varnes (1996) | text | 3 classes (earth, debris, rock) |
| ROUGH_VOL | Rough volume of the displaced material | numeric | expressed in $m^3$ |
| WATER_TYPE | Type of watershed in which the landslide-affected site occurs | text | 3 classes (coastal, endorheic, inland) |
| WATER_NAME | Name of the watershed in which the landslide-affected site occurs | text | free |
| GEOM_CONT | Geomorphological context in which the landslide-affected site occurs | text | 3 classes (inner slope, coastal slope, quarry) |
| ELEVATION | Elevation of the landslide-affected site | numeric | expressed in m a.s.l. (2-682) |
| SLOPE | Slope of the landslide-affected site | numeric | expressed in angular degree (4-77) |
| ASPECT | Aspect of the landslide-affected site | numeric | expressed in angular degree (1-360) |
| UNIT_W | Unit weight of the involved lithological unit (average) | numeric | expressed in $kN/m^3$ (13.0-27.25) |
| FRICTION_A | Friction angle of the involved lithological unit (average) | numeric | expressed in angular degree (25-40) |
| COHESION | Cohesion of the involved lithological unit (average) | numeric | expressed in kPa (1-800) |
| Vs30 | Average shear-wave velocity in the first 30 m of the site stratigraphic succession | numeric | expressed in m/s (150-1275) |
| AVG_THICK | Average thickness of the pyroclastic cover | numeric | expressed in cm (7-150) |
| HAZARD | Hazard level related to the landslide-affected site, as evaluated by the local Basin Authority | numeric | 4 classes (1, 2, 3, 4) |
| SUSC | Susceptibility level related to the landslide-affected site, as evaluated by the local Basin Authority | numeric | 3 classes (1, 2, 3) |
| DAMAGE | Type of damage | text | free |
| INJURED | People injured | numeric | number (1-7) |
| FATAL | People killed | numeric | number (1-25) |
| CAUSE | Landslide predisposing or triggering factor | text | 7 classes (digging, dumping, earthquake, mining, rainfall, water leak, wildfire) |

## 4 Results

The CAFLAG geodatabase encompasses 2302 landslides occurred throughout the continental, coastal and insular sectors of the Campi Flegrei, during the 1828 - 2017 time span (Fig. 3). Landslides characterized by high accurate position are 2122 (about 92 % of the total). Those with a temporal accuracy ranging from moderate to high are 482, corresponding to about 21

% of the total. Statistics presented in the following sections describe some general features of the collected data, which are useful to characterize the landslide phenomena in the area.

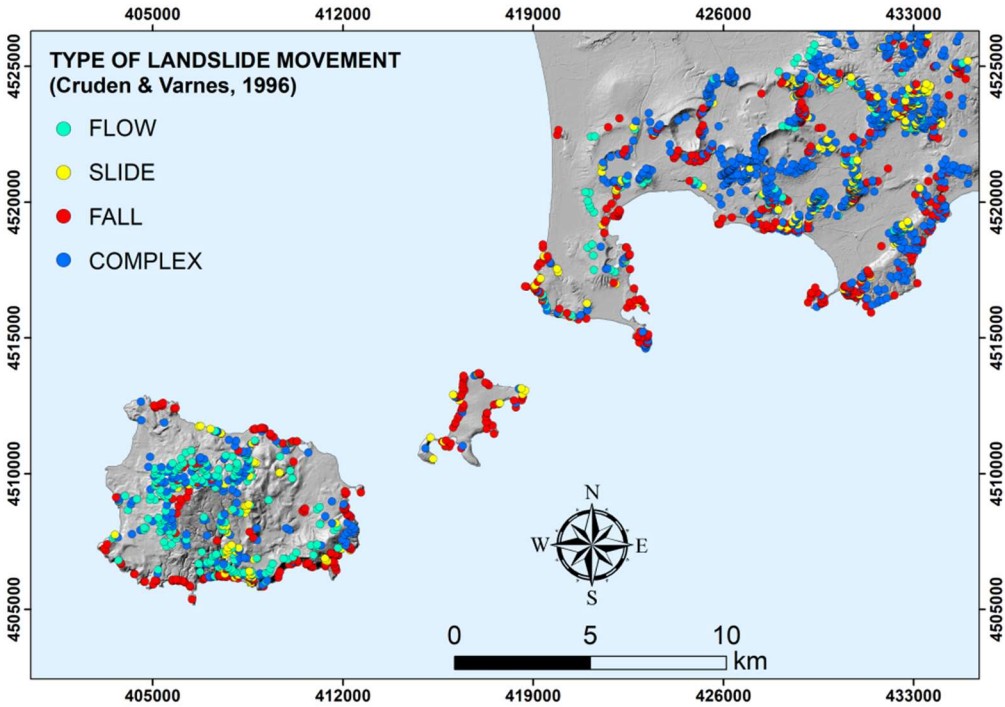

Figure 3: Map of the cataloged landslides, classified according to the type of movement.

## 4.1. Landslide general features

In this section, statistical distributions of the landslide movement types, displaced material, and displaced volume are described. Data in Fig. 4 highlight that most of the cataloged landslides are characterized by a complex movement (1115). Among them, specific information about the failure mechanism is available for 343 complex landslides only (15%), whereas no indication is available for the others 772 (34 %). The most recurrent movement type among the non-complex landslides is fall (621 events - 25%), which is slightly more frequent than slide and flow types (13% each).

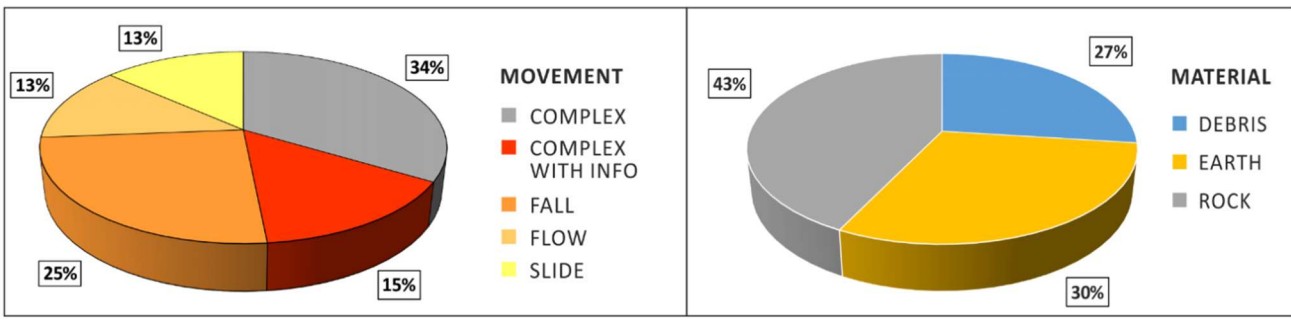

Figure 4: Landslide events classified according to the movement type, on the left, and displaced material, on the right.

Examples of rock falls are showed in Fig. 5. These failures are quite frequent both along the coastal and inner tuffaceous cliffs, determining episodic and localized cliff retreats that pose a serious risk for buildings and people living nearby to the cliff tops. These are able to displace huge rock masses, mobilizing up to thousands cubic meters of volcaniclastic rocks.

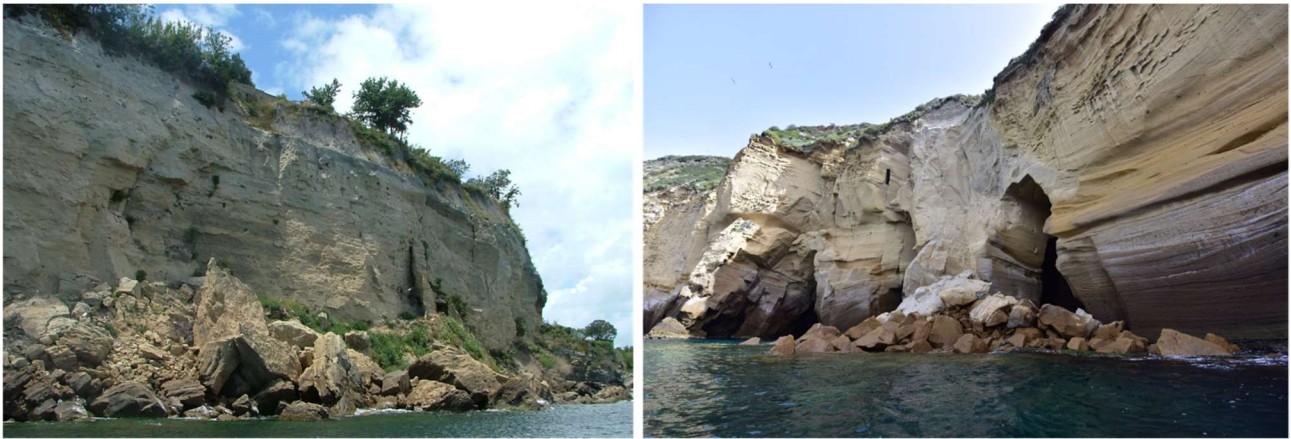

**Figure 5: Examples of rock fall affecting the Campi Flegrei coastal cliffs: Punta Miseno (2015) on the left, and Trentaremi - Capo**
**Posillipo (2017) on the right (both sites are indicated in Fig. 1). (Image credits: Alessandro Fedele, INGV).**

The Fig. 6 shows instead a series of rainfall-induced shallow landslides displacing unconsolidated pyroclastic deposits and soils. Images refer to two different events affecting the Monte Vezzi (Ischia island) in April 2006, and the coastal slope of Monte di Procida in 2010. Rocky cliffs like those represented in Fig. 5 result the most affected by landslides (43 %), whereas
slopes covered by soils and fine pyroclastic deposits (earth) (see examples in Fig. 6) result affected by 30 % of the events. Coarser pyroclastic deposits or ancient landslide and debris deposits are those less involved by mass movements (27 %) (Fig. 4).

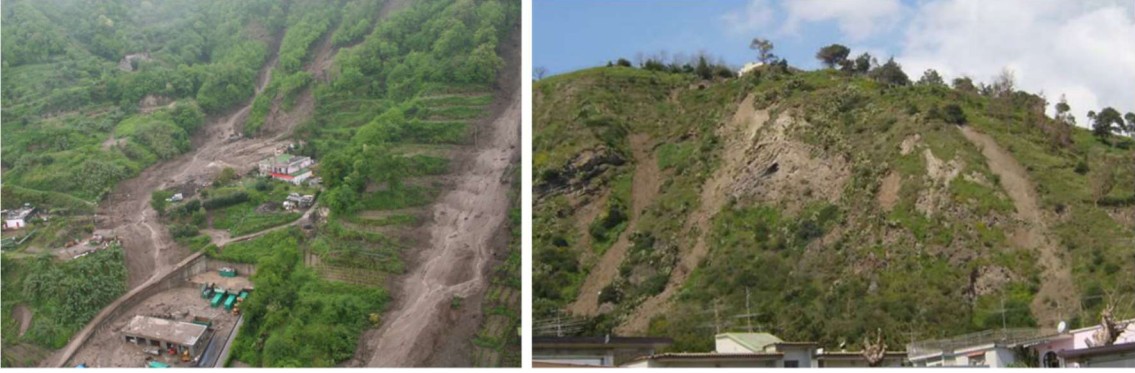

**Figure 6: Examples of flow-like mass movements at Monte Vezzi in the Ischia island (2006), on the left, and at Miliscola - Monte di**
**Procida (2010), on the right (both sites are indicated in Fig. 1). (Image credits: Nucleo elicotteri Vigili del Fuoco Salerno; Prof. Paola**
**Romano).**

Volumetric data are available for 277 landslides (Fig. 7), corresponding to about 12 % of the whole dataset. Values range from 1 to 7.500 $m^3$ with a mean of about 123 $m^3$ per event. About 40 % of the 277 landslides displaced a volume lower or equal to 10 $m^3$, and 23 % a volume higher or equal to 100 $m^3$. As shown in Fig. 7, the sharp break in the curve slope located at around 10 $m^3$ highlights that landslides with volumes below this value are strongly underestimated. This is likely due to the missed reporting of minor landslides, which often do not cause damage or casualties.

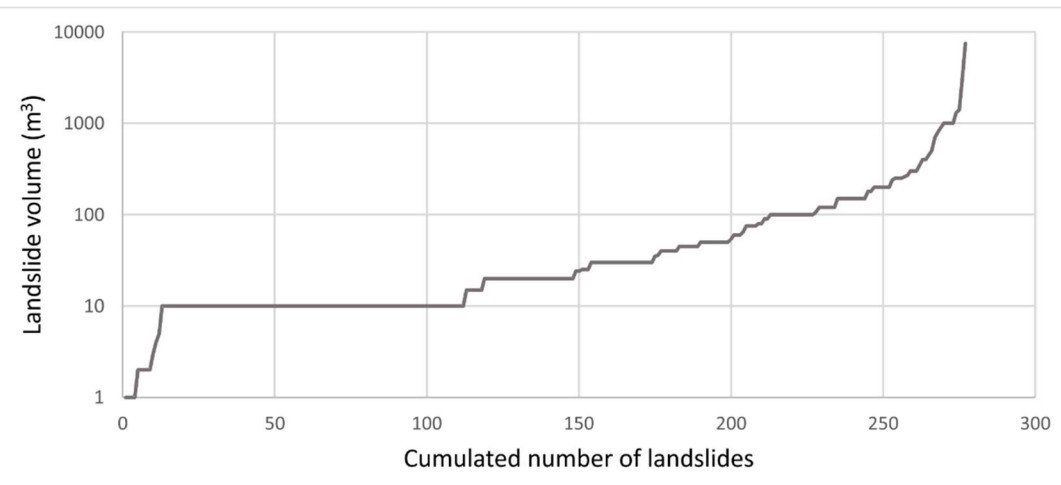

**Figure 7: Cumulated frequency distribution of the 277 landslides with volumetric data.**

## 4.2. Temporal distribution

The year of occurrence is known for 517 landslides only, with monthly information available for 482 of them. Incompleteness of historical information in the first half of the analyzed period results in a yearly trend (Fig. 8A) characterized by a misleading linear increase in the number of landslides. On the other hand, peaks in the years 1986, 1997, 2005 (i.e. 50, 91, 70 events, respectively), that exceed significantly the average value of 2.4 events per year, can be retained realistic because occurred in the period (i.e., between 1985 and 2017) with highest availability of documentary sources. In this period, the annual frequency assumes an almost constant average value of 9.9 events per year. Regarding the seasonality, most of the events occurred in the winter season, and specifically in the months from January to March, even if a number of events resulted also in the late summer-fall months (Fig. 8B).

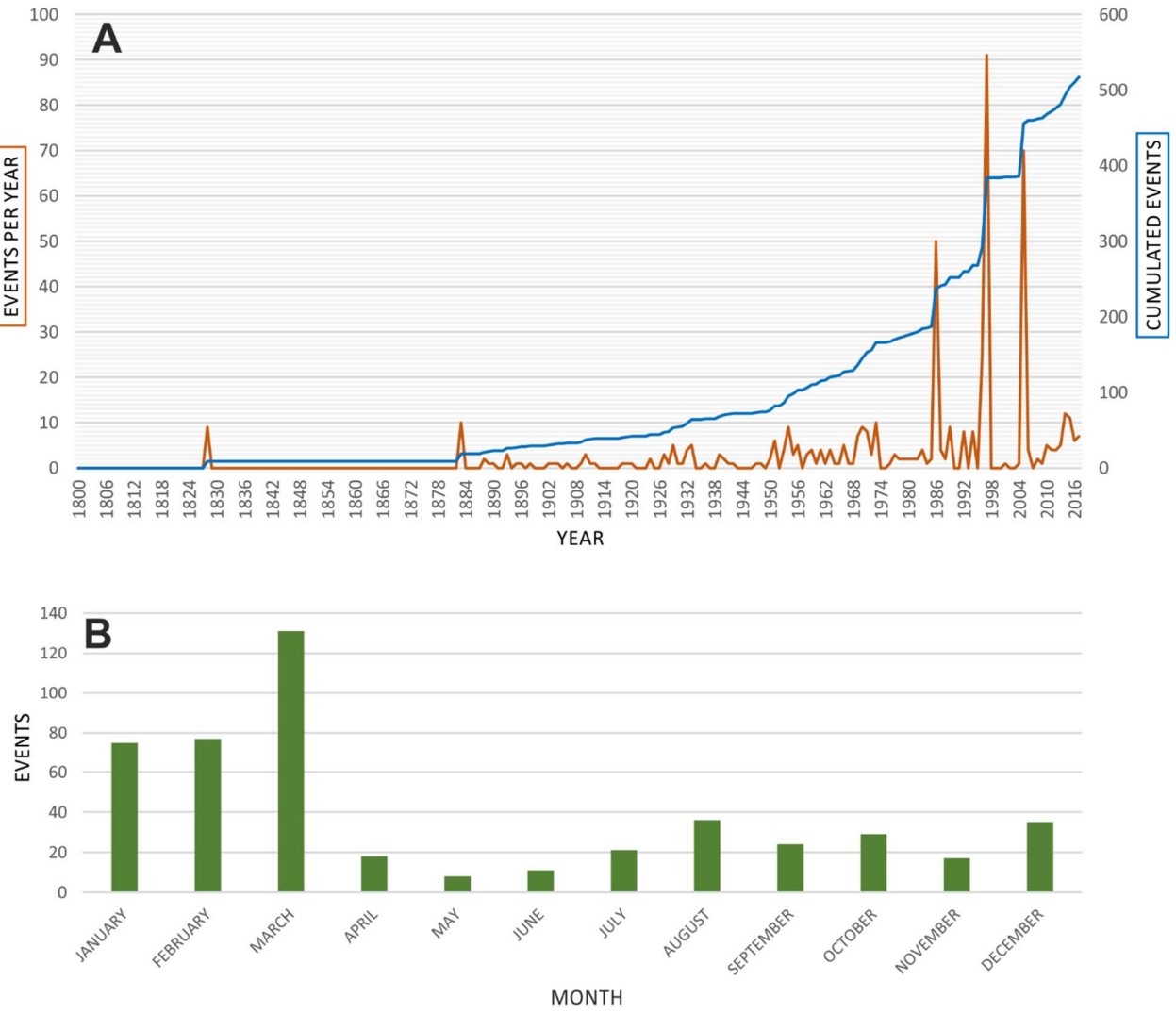

**Figure 8: Yearly (A) and monthly (B) distribution of the landslide events characterized by temporal information.**

### 4.3. Deadly impact and societal risk

According to the information collected in the CAFLAG geodatabase, 127 people lost their life as consequence of 53 fatal landslides occurred in 105 years (1912-2017) with an average rate of 1.2 fatalities per year. The temporal distribution of fatalities is represented in Fig. 9; here, the cumulative curve has a stepped shape in a broader concave (downward) trend, and its slope increases in correspondence of significant events in terms of fatalities, such as the landslide of 1948, corresponding to the deadliest within the analyzed time period (25 fatalities). Since the early 1980s, the average frequency of fatalities per year decreased from 1.7 (related to the time span 1910-1980) to 0.3, as highlighted by a gentler slope of the cumulative curve. This reduction in the trend, also documented at national scale (Rossi et al., 2019; Esposito et al., 2023), can be due to the

increased resilience of local communities based on new technologies and prevention measures, as well as on a higher awareness about the risk posed by geo-hydrological processes.

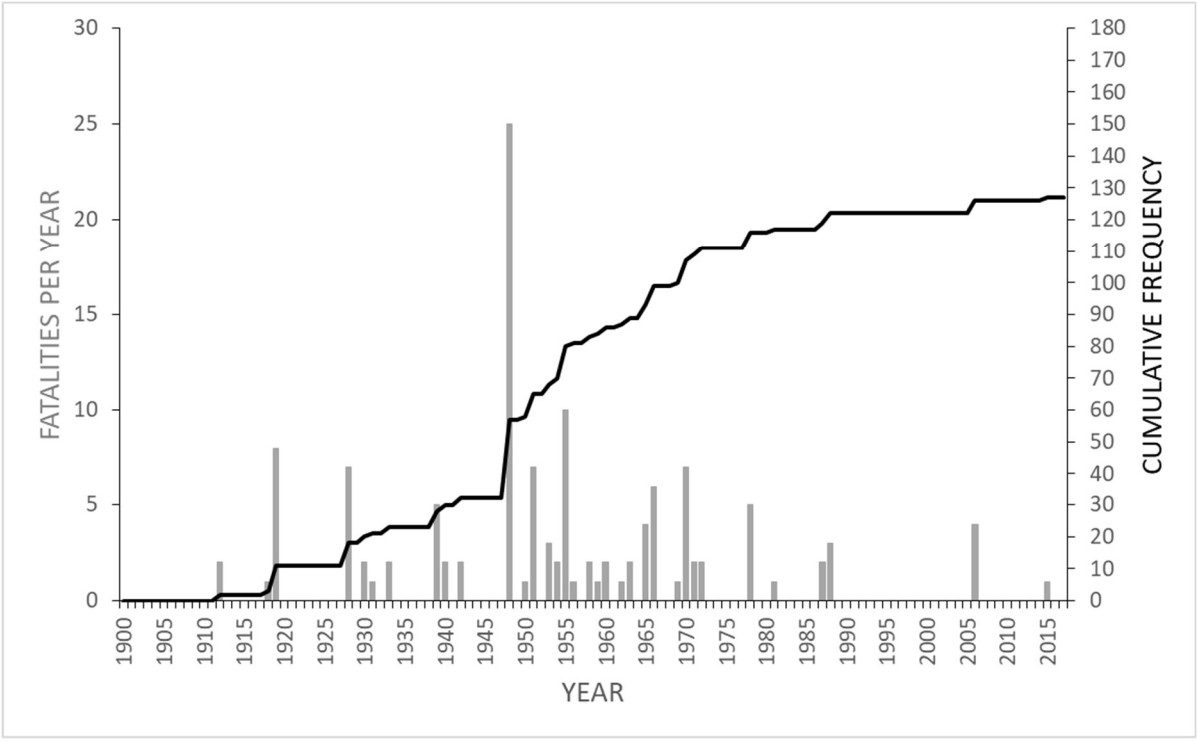

**Figure 9: Statistics of fatalities caused by landslides in the analyzed time span.**


In order to analyze the societal risk posed by landslides in the study area since 1912, the F-N curve was developed for both continental and insular sectors, and compared with the F-N curve realized by Cascini et al. (2008) for a larger area of the Campania region characterized by pyroclastic deposits covering a volcanic substratum (lava and tuff). This curve represents the cumulative probability per year (F) that landslides will cause N or more fatalities versus the number of fatalities resulting

from landslides (Fell and Hartford, 1997). The F-N curve in Fig. 10 is characterized by a relatively high slope owing to the great number of incident data which refer to landslide events resulting in no more than 25 fatalities, and by the lack of more ancient data than 1912 not considered in this study. The curve obtained for the study area (Fig. 10) shows that the societal risk posed by landslides is very high when compared to that estimated by Cascini et al. (2008) for the whole volcanic areas of the Campania region. The two curves show a similar behavior until N=10. After that point, the curve of Cascini et al. (2008) goes

down towards lower F values controlled by fatalities in the interval 9<N<25, which do not occur in this study. This result can be of particular concern for public authorities in charge of landslide risk mitigation by emergency plans and warning systems, also in order to define acceptable and tolerable landslide risk thresholds by society. Generally, as recognized for the whole

Campania territory (Cascini et al., 2008), anthropogenic factors represent the most important triggering cause of landslides associated to no more than 3 fatalities, while critical rainfall is able to trigger also landslides with worse consequences in terms of fatalities (e.g. landslides of 26 November 2022 at Casamicciola - 12 fatalities).

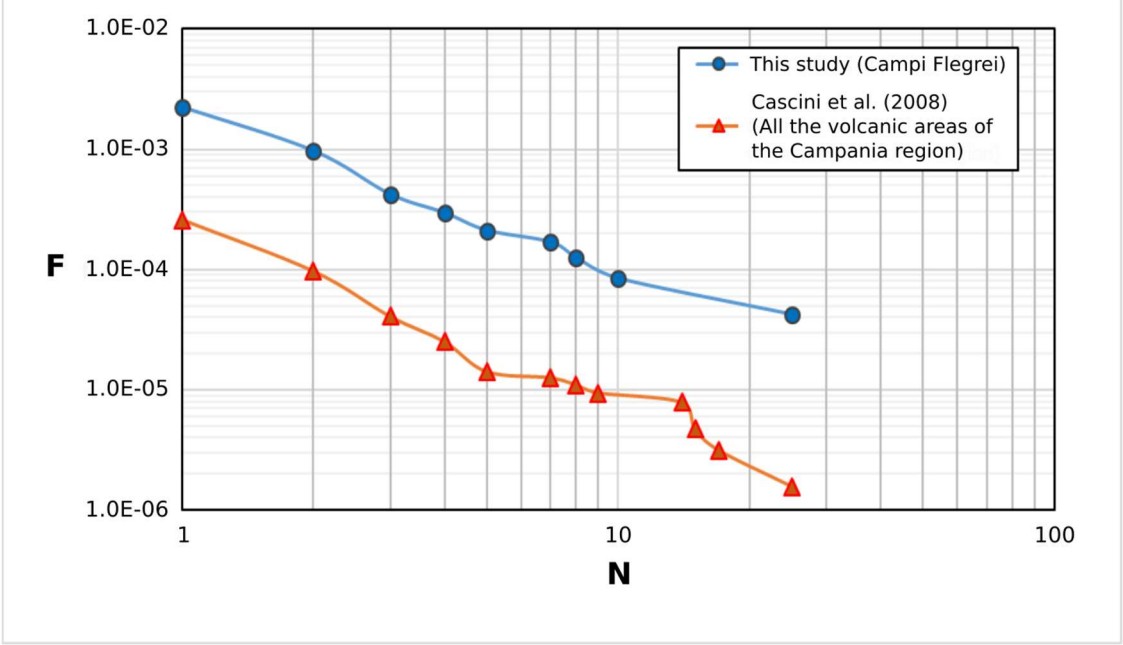

**Figure 10: F-N curve of the Campi Flegrei study area compared with the F-N curve developed by Cascini et al. (2008) for a larger area of the Campania region, encompassing Campi Flegrei, Vesuvius and Roccamonfina volcanic sectors. F is the annual frequency of events causing N or more fatalities. The two curves have been normalized for the areal extension of the investigated areas (1741 km² for the whole volcanic areas of the region and 230 km² for the Campi Flegrei study area).**

### 4.4. Geomorphic and engineering geological properties of the affected hillslopes

Statistics related to the 2122 landslides characterized by an accurate location (high) show that 53 % of them occurred in steep terrains characterized by slope angles ranging from 30 to 77 degrees, whereas 47 % in more gentle slopes characterized by slope angles lower than 30 degrees. The highest absolute frequency of events results in the slope class from 30 to 40 degrees (22 %); the highest densities (events/km²) occur in the classes 50°-60° (92 events/km²) and 60°-70° (97 events/ km²) (Fig. 11). Aspect data in Fig. 11 show the highest density of landslides along slopes exposed towards the North-West direction (i.e. 315°-360°), and the highest absolute frequency in the aspect class from 225° to 270°. A lower frequency of events results instead for slopes exposed towards East. This can be linked to the direction from which the rainstorms associated to low-pressure systems usually came from, such as the Tyrrhenian Sea (Fig. 1) located to the West of the study area (Saviano et al., 2019; Fortelli et al., 2019).

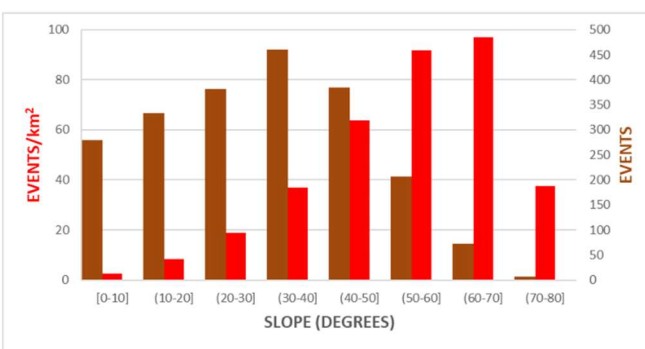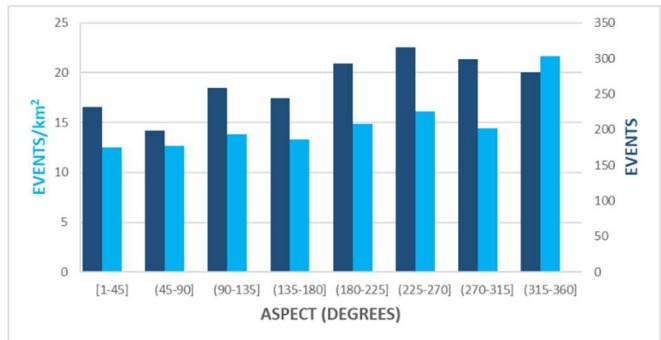

Figure 11: Distribution of the landslide events according to the slope gradient and aspect of the affected hillslopes.

Most of the landslides occurred along the inner hillslopes (78 %), about 21 % affected coastal cliffs, and 1 % occurred along quarry cliffs. On the other hand, by considering the relative area of these three geomorphological contexts, the highest concentration of landslides can be found along the coastal sector (Fig. 12), confirming the strongly active coastal dynamics in the study area, as deduced also from the coastal cliff retreat rates (Esposito et al., 2018a).

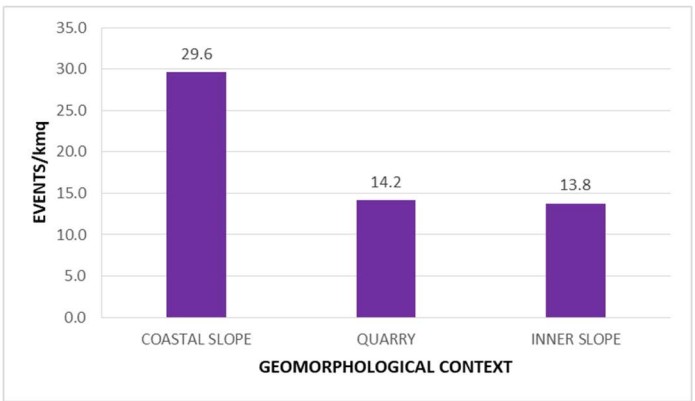

Figure 12: Distribution of the landslide events according to the geomorphological context.

Statistics of the geotechnical properties characterizing lithologies involved in 2106 mass movements reveal a concentration of records (92 %) in correspondence of average unit weights typical of volcaniclastic deposits, in a range from 13 to 17 kN/m$^3$ (Fig. 13A). A few events (8 %) displaced lithologies characterized by higher values, corresponding to lavas and other solid rocks outcropping in the area. Histogram of the friction angle highlights values typical of pyroclastic rocks (Caccavale et al, 2017 and reference therein) ranging from 25° to 40°, depending on the degree of density and mineral composition, with the highest frequency in the ranges from 25° to 31° (Fig. 13B). At the same time, cohesion also concentrates in the typical interval of weakly welded pyroclastic deposits, and specifically below 50 kPa, whereas higher values refer to the most lithified rocks (Fig. 13C).

Statistical distribution of the Vs30 parameter indicates that 73 % of landslides (out of 2078) affected lithologies characterized by low Vs30 values, ranging from 100 to 400 m/s (Fig. 13D), which can be explained by weak geotechnical properties of the pyroclastic deposits documented in the area (e.g., Rowley et al., 2021; Albano et al., 2018; Caccavale et al., 2017).

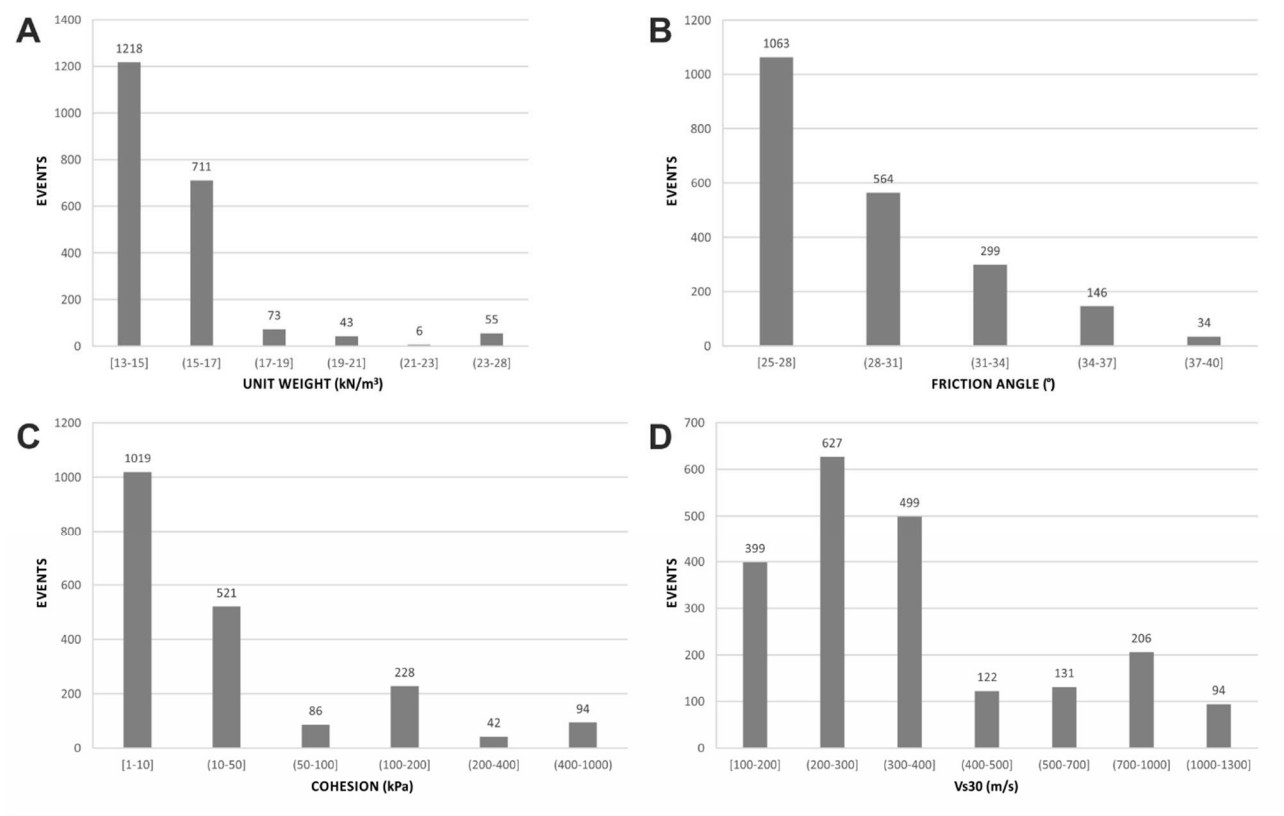

325

Figure 13: Distribution of the landslide events according to the unit weight (A), friction angle (B), cohesion (C) and Vs30 (D) of the displaced lithologies.

## 4.5. Limitations of the CAFLAG geodatabase

330 Historical landslides catalogues are commonly affected by incompleteness and biases resulting from the type of sources used to collect information (Pereira et al., 2014). These limitations are usually of unknown effect or seriousness, and there are no general data analysis rules for eliminating them (Ibsen and Brunsden, 1996). The analysis of the completeness of historical catalogues of natural events is almost always difficult (Rossi et al. 2010) and, in case of landslides, it depends on the assessment of the fraction of mass movements in an inventory compared to the real number of events in the study area (Guzzetti et al.,

335 2012). The CAFLAG geodatabase refers to a relatively small geographical area with respect to other published regional or

national historical landslide databases, as those listed by Bíl et al. (2021). On one side, this reduces the biases in the spatial accuracy and completeness of the geodatabase. In fact, 92% (2122) out of the 2302 records are characterized by high spatial accuracy, highlighting an adequate localization of the recorded landslides. At the same time, the spatial completeness can be considered relatively high since many small and isolated events are also reported. In addition, indirect evidence of the high completeness of the CAFLAG geodatabase can be derived from a comparison with other databases covering comparable time ranges, and spatial extents even larger. For example, Devoli et al. (2007) cataloged 132 historical landslides occurred in the Nicaragua country, in the time period 1826 - 1988 (162 years); Elliott and Kirschbaum (2007) realized a preliminary database covering the Utah State (USA), encompassing 356 landslide occurrences from 1850 to 1978 (128 years); Laprade et al. (2000) reported 1326 landslides which occurred between January 1890 and June 1999 (109 years); Pereira et al. (2014) reported 628 landslides in northern Portugal in the period 1900 – 2010 (110 years). All these databases report a lower number of records compared to those (n. 2302) within the CAFLAG geodatabase. Besides this, it is worth noting that the completeness of the CAFLAG geodatabase is certainly poor in those areas where the population density is low, such as in correspondence of the coastal cliffs or close the craters of volcanic edifices (Fig. 3).

An evident bias affects the temporal information. In fact, landslides with a high temporal accuracy (i.e. dd/mm/yyyy) are 344 (15%) only, whereas most are characterized by low accuracy (79%), and the remaining part by moderate accuracy (6%). Such a bias depends on the usage of geomorphological inventories developed mostly by means of visual interpretation of aerial photographs (e.g., IFFI, inventory map of the AdB, CARG project) which do not allow to determine accurate temporal information on the occurrence of landslides (Trigila et al., 2010). In addition, the steady growth of the number of reported landslides over time, shown in Fig. 8A, is ostensibly controlled by an increased availability of documental sources rather than a real rise in mass movements, as observed at both national scale (Guzzetti and Tonelli, 2004) and in other Italian regions (Salvati et al., 2009; Piacentini et al., 2018). On the other hand, peaks recorded in the years 1986, 1997 and 2005 characterized by 50, 91, and 70 events respectively are likely due to a particularly high landslide activity and require an in-depth analysis. Information on the displaced volumes is available for 12% of the events, whereas the type of damage is reported for 41 events only, representing further limitations in the magnitude information reported within the CAFLAG geodatabase.

## 5 Data availability

The CAmpi Flegrei LAndslide Geodatabase (CAFLAG) is freely available in the 4TU.ResearchData repository at https://doi.org/10.4121/14440757.v2 (Esposito and Matano, 2021).

## 6 Summary and conclusions

Historical catalogues represent a unique information to inspect the extent, types, pattern, recurrence and statistics of slope failures in a given territory (Guzzetti et al., 2012). In volcanic landscapes, these catalogues are even more necessary to reach

a comprehensive knowledge about the role played by landslides in the geomorphic evolution of these settings, and on processes that lead to slope failures and control their spatial variations (Yano et al., 2019). As highlighted by data reported within the CAFLAG geodatabase, weak geotechnical properties of the volcanic lithotypes (Fig. 13), coupled with a rough topography play a relevant role in predisposing volcanic hillslopes to mass wasting processes. This should be demonstrated also with numerical analyses by exploiting the twofold contributions provided by the CAFLAG geodatabase, consisting of data related to both landslide processes and geo-environmental properties of the affected sites which can be exposed to new instabilities. These two types of information coincide in fact with inputs required by statistically-based landslide susceptibility models, namely "dependent" and "explanatory" variables respectively (Reichenbach et al., 2018). Therefore, data of the CAFLAG geodatabase may be aimed at calibrating or validating statistical models resulting in landslide susceptibility and hazard maps which, in the study area, may complement the official hazard zonation.

The high spatial accuracy of the cataloged records, detailed information on typologies, morphometric and geotechnical properties of the affected sites provide hence a valuable opportunity to investigate the occurrence of landslides in relation to other geo-environmental predisposing or triggering factors. For instance, analyses described in the previous sections highlight that mass wasting processes concentrated along the coastal sector. Collapses of the coastal cliffs can be considered the most important agents leading to cliff retreat in the area, since these are able to remove large volumes of rocks in short times (Esposito et al., 2017, 2018a). The CAFLAG geodatabase could be exploited hence to assess susceptibility of the rocky coastline against marine erosion, or risk conditions affecting the widespread anthropic settlements and people there located.

Statistics show that rocky slopes formed by lithified volcanic rocks, such as tuff or ignimbrite, were mostly affected by rock falls. In the coastal area, this was documented by Ducci and Napolitano (1991), who analyzed the cliff failures along the Procida island coastline observing that all of the identified collapses were rock falls and topples involving tuffaceous formations. Similar results were achieved by Del Prete and Mele (1999) for the cliffs of the Ischia island. In the inland areas, however, many landslides affected the walls of old and abandoned quarries or digging front (Calcaterra et al., 2007). Discontinuity systems characterizing rock masses play a fundamental role in predisposing rocky slopes to failure processes, as well as in controlling the volume of the mobilized rocks (Matano et al., 2016). In the study area, discontinuities consist of faults, joints and fractures associated to volcano-tectonic processes (Vitale and Isaia, 2014). Other failure-predisposing factors include geotechnical properties (e.g. low cohesion in Fig. 13) and weathering processes typical of the Flegrean area, such as the zeolitization and argillification of pyroclastic rocks. Water circulation, earthquakes, sea wave action, or anthropic vibrations and excavations can be considered among the main triggering factors. Besides the dominance of landslides involving solid rocks, the shallow sliding of loose pyroclastic deposits (i.e. pumices, scoria, ashes and lapilli) and soils covering the lithified formations is also frequent. Commonly, these failures start as translational slide and evolve into debris or hyperconcentrated flows within gullies, or debris avalanches along the steep open slopes (Fig. 6) (Calcaterra et al., 2003a). Intense rainfall represents the triggering factor of the initial slide. Elevated slope angles and geotechnical properties of the involved material (Fig. 13) lead to a rapid or very rapid evolution in flow processes, as observed in the recent event of Casamicciola.

Many events affected hillslopes exposed towards west, north and south. This may be explained by the frequent impact of these hillslopes with convective cells and low-pressures systems coming from the Tyrrhenian Sea (Saviano et al., 2019). These meteorological structures are able to release high amounts of rainfall causing also flash flood processes (e.g. Esposito et al., 2015, 2018b), and are often associated with strong winds and storm surges damaging coastal infrastructures, as documented in the city of Naples in 2020 (Fortelli et al., 2021).

Another relevant finding of the current analysis is that in the analyzed time period, 53 landslide events caused 127 fatalities with an average rate of 1.2 fatality per year (Fig. 9). Most of the people died in the city of Naples, hit by rocky blocks detached from steep tuffaceous slopes; others died because of flow-like mass movements, mostly recurrent in the Ischia island (e.g. Santo et al., 2012). The intense urbanization of the area developed after the Second World War determined an increase of fatal landslides (Fig. 9) and risk conditions in a way that, as highlighted also by the F-N curve (Fig. 10), landslides in the area

represent a relevant societal risk. Data contained in the CAFLAG geodatabase may be thus exploited for applications aimed at planning countermeasures for the landslide risk reduction.

**Author contributions.** GE and FM collected and processed the data. GE prepared the first draft of the paper. FM edited and integrated the text at several stages, supervising the research activity.

**Competing interests.** The authors declare that they have no conflict of interest.

**Acknowledgements.** A first draft of the geodatabase was developed within the Research Project PON MONICA funded by the Italian Ministry of Education, University and Research (grant number PON01_01525 - Coordinator of the CNR-IAMC Research unit: Dr. Marco Sacchi; National Coordinator: Dr. Giuseppe De Natale - INGV). We are grateful to the Editor and four anonymous reviewers for their constructive comments that helped us to improve the content, quality, and readability of
the work.

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
