# Peer review of "A geodatabase of historical landslide events occurred in the highly urbanized volcanic area of Campi Flegrei, Italy"

_Earth System Science Data, 2022_

## Referee Comment (RC1)

In your study entitled 'A geodatabase of historical landslide events occurred in the highly urbanized volcanic area of Campi Flegrei, Italy' is an interesting work that tries to share data on historical landslide events. The data are deposited at 4TU.ResearchData in the form of shpfile and kml. The data are relevant for a community of landslide hazard research, and principally this is a work that justifies publication in ESSD. However, there is some weakness in the way the data are organized and presented. After carefully reading the MS and exploring the attached dataset, I concluded that the way this dataset is presented is insufficient for ESSD in its present form, following the main comments and the specific comments listed below.

**Main overall concerns:**

The MS does not fit well into the scope of ESSD in its present form. It lacks to interconnect the data it shares and to show how it is valuable in relation to the Earth's system.

The title says '…historical landslide events…' yet It seems to me that sections 4.1 and 4.3 are the ones that truly reflect the title of the MS. Sadly, the other valuable information that was presented in the earlier sections is not tied into the main logical thread of the MS, nor it is demonstrated how vital those may be to be used with historical landslide events.

The presentation of the dataset is poor. I am missing at least one table/figure that really gives an overview on the whole dataset that is presented in the MS from the variables side. After all that is in the focus according to the title. The information provided in Sect. 2 is relevant, yet insufficient. The reader must get an overview on the dataset that is presented, before any detail on the measurements is discussed.

The study lacks to show how the data it shares is relevant in geological hazard research as one would expect based on the introduction.

**Specific comments:**

**Abstract**

The penultimate sentence tries to describe how the dataset could be utilized. This should be much more elaborated on and should be one of the main messages of the abstract.

Line 9-10: It would be better to give concrete examples.

Line 14: Why is it from 1828 and not earlier? It's already 2022. Why is there no update? Where does the data from? Is it reliable? How to check the quality?

Line 20: What do you want to say? What do we get out of this?

Line 21-25: I think the purpose of collecting this data should be briefly described in the abstract, and what can we do with this data? Instead of just saying landslides have a significant impact on humans, we need to use the knowledge to deal with them, so what do we get from your data set? Is it just simple data?

Introduction

The Introduction seems as if it was written to another paper. The introduction correctly addresses issues that the dataset at hand could be used to solve. However, when the dataset is presented its values and possible applications are not presented in light of the Introduction.

Line 38: What additional risks?

Line 41: The importance of the region should be explained internationally, otherwise, there is no comparison, only regional characteristics cannot be promoted globally.

Line 52-53: And then what? What are you trying to say? What can we learn from these deaths?

Figure 1: Its global location is unknown. It is recommended to add latitude and longitude to add elevation data, now do not know the region's topography, topographic conditions, and the water system's distribution.

Data and methods

Line 111: Why do you choose this time period? What about other times?

Line 127: Why is there no comparison with other categories? Would it be accidental to use this category? Then how do we control the quality of data?

**Line 138:** I don't think this is very good statistical software. Why not use R, Python, and Matlab for analysis? How is your significance tested in EXCEL? This leads me to question the availability of data quality.

**Table 1:** It is suggested that the author upload the disaster on that day and year in the form of documents to facilitate readers' visualization.

**Study area**

**I miss a description of the area's water system, vegetation, soil, and population distribution, as described in the abstract.**

**Line 175:** Lack of references.

**Results**

**Figure 3:** Can we get more information by discussing the seasons?

**Figure 4:** I don't see any buildings. Instead, I think it was taken far away from where people live.

**Figure 5:** It is recommended to take the photo from the same angle, choosing the same reference object for easy comparison.

**Line 210-213:** What do we learn from these statistics?

**Line 249:** From what? On what basis?

**Section 4.3:** So, what useful information can we draw from this? In addition, the content order of the result part needs to be modified, and there is no logic among the four results.

**Figure 10:** I suggest adding a trend line for intuitive analysis.

**Line 277:** What do we get from the F-N curve?

**Line 279:** Why choose this time period? It's not in the scope of your dataset.

**Summary and conclusions**

**Line 293:** This should be in the introduction.

**Line 312:** How do you reach this conclusion? It is suggested to add the circulation system and the distribution map of sea level pressure in the article.

**Line 317:** How is this conclusion known? Urban maps of people killed by landslides should be added.

---

## Referee Comment (RC3)

[referee-annotated manuscript omitted]

---

## Author Response (AR1)

**Point-by-point response to the Reviewer #1**

**General comments**

**Reviewer comment:** In your study entitled 'A geodatabase of historical landslide events occurred in the highly urbanized volcanic area of Campi Flegrei, Italy' is an interesting work that tries to share data on historical landslide events. The data are deposited at 4TU.ResearchData in the form of shpfile and kml. The data are relevant for a community of landslide hazard research, and principally this is a work that justifies publication in ESSD. However, there is some weakness in the way the data are organized and presented. After carefully reading the MS and exploring the attached dataset, I concluded that the way this dataset is presented is insufficient for ESSD in its present form, following the main comments and the specific comments listed below.

**Author response:** We are grateful to the Reviewer #1 for his/her comments and suggestions aimed at improving our manuscript. Point-by-point responses to all the comments are outlined below.

**Reviewer comment:** The MS does not fit well into the scope of ESSD in its present form. It lacks to interconnect the data it shares and to show how it is valuable in relation to the Earth's system.

**Author response:** Thanks to the comments and suggestions provided by the two Reviewers, we have deeply revised the present form of the manuscript for making it suitable to be published, hopefully, in ESSD.

**Reviewer comment:** The title says '…historical landslide events…' yet It seems to me that sections 4.1 and 4.3 are the ones that truly reflect the title of the MS. Sadly, the other valuable information that was presented in the earlier sections is not tied into the main logical thread of the MS, nor it is demonstrated how vital those may be to be used with historical landslide events.

**Author response:** The title in the current form provides information on the main topic addressed by the article, and responds to the ESSD request, such as to be concise and informative. We suggest to do not modify it, and to do not extend its length. Information associated to the landslide events are strictly related to them, since they consist of data explaining the impact and relationships with geological and geomorphological properties of the affected sites (i.e. predisposing factors). In the light of this, the term "historical landslide events" in the title should be considered in its broadest meaning.

**Reviewer comment:** The presentation of the dataset is poor. I am missing at least one table/figure that really gives an overview on the whole dataset that is presented in the MS from the variables side. After all that is in the focus according to the title. The information provided in Sect. 2 is relevant, yet insufficient. The reader must get an overview on the dataset that is presented, before any detail on the measurements is discussed.

**Author response:** It seems to us that all the Reviewer's requests included in this comment can be satisfied with contents already available within the Section 2 of the article, including a detailed table explaining all the attributes associated to each record.

**Reviewer comment:** The study lacks to show how the data it shares is relevant in geological hazard research as one would expect based on the introduction.

**Author response:** In the revised version of the manuscript, we have added more information on how the CAFLAG geodatabase could be exploited by both broad and local geoscience research community. In particular, within the "summary and conclusions" section we have pointed out that the CAFLAG dataset could be used to: i) enhance scientific knowledge about the role played by landslides in the geomorphic evolution of volcanic landscapes, and on processes that lead to slope failures and control their spatial variations; ii) calibrate and validate statistically-based landslide susceptibility models; iii) assess risk conditions and support the planning of mitigation measures.

**Specific comments**

**Abstract**

**Reviewer comment:** The penultimate sentence tries to describe how the dataset could be utilized. This should be much more elaborated on and should be one of the main messages of the abstract.

**Author response:** Please, see the response to the previous comment. In addition, we would like to remark that we have provided some suggestions on the possible uses of this geodatabase according to our knowledge and experience, highlighting also the related potentialities and limitations in the new section 4.5. Besides this, we are confident that this geodatabase can be exploited by the geoscience research community for more possible uses and in further different ways with respect to the proposed ones.
We suggest to do not focus the abstract on the possible use of the geodatabase, since this theme has been already remarked in the first part of the abstract (lines 2-3).

**Line 9-10:** It would be better to give concrete examples.

**Author response:** The highlighted sentence aims to introduce the Campi Flegrei case study, representing a concrete example of "urban settlements exposed to multi-hazard conditions", since both volcanic, seismic, flash flood, landslides, and densely population occur.

**Line 14:** Why is it from 1828 and not earlier? It's already 2022. Why is there no update? Where does the data from? Is it reliable? How to check the quality?

**Author response:** The CAFLAG geodatabase includes events occurred since 1828 and not earlier because previous data were not considered reliable for further scientific analyses, that is the scope of the geodatabase and hence of this article. There are no updates after 2017 because, in the successive years, the research efforts were concentrated on coastal landslides, as documented by works cited also in this article.
The data sources are described in detail within the section 2.1.
Most of the used data come from international scientific publications and official landslide databases and inventories developed by Italian research institutions and public agencies (ISPRA, CNR, Basin Authorities), guaranteeing proper and independent validation procedures leading to high quality and reliable information. Data related to the most recent events (2013-2018) collected from newspapers and websites were validated with field observations.

**Line 20:** What do you want to say? What do we get out of this?

**Author response:** In accordance with suggestions of the Reviewer #3, we have removed this sentence.

**Line 21-25:** I think the purpose of collecting this data should be briefly described in the abstract, and what can we do with this data? Instead of just saying landslides have a significant impact on humans, we need to use the knowledge to deal with them, so what do we get from your data set? Is it just simple data?

**Author response:** In accordance with the Reviewer's suggestion, we have modified this part of the abstract by providing some brief suggestions about the possible use of landslide fatality data.

**Introduction**

**Reviewer comment:** The Introduction seems as if it was written to another paper. The introduction correctly addresses issues that the dataset at hand could be used to solve. However, when the dataset is presented its values and possible applications are not presented in light of the Introduction.

**Author response:** The main purpose of the current manuscript is to describe research data included within the CAFLAG geodatabase, as recommended within the ESSD aims and scope section for data description papers. In addition, instructions for authors clarify that "extensive interpretations of data remain outside the scope of this data journal". The possible applications that, according to the Reviewer, we should suggest into the article are not recommended to the authors. Therefore, it seems to us that such suggestions represent a kind of interpretation, since each researcher can use datasets according to its knowledge and necessity, and there is not a prefixed scheme to use landslide datasets that we have to remark. Besides this, we agree with the Reviewer that some recommendations can be provided, by taking also into account the quality of data. For this reason, we have improved the final part of the manuscript providing more information on these issues.

**Line 38:** What additional risks?

**Author response:** We mean that landslides pose a risk that is additional to those posed by volcanic and seismic processes in the area. We have modified the sentence to be more clear.

**Line 41:** The importance of the region should be explained internationally, otherwise, there is no comparison, only regional characteristics cannot be promoted globally.

**Author response:** As stated into the text, the Campi Flegrei volcanic area corresponds to an active volcanic caldera considered among those with the highest volcanic risk in the world. This is well known within the scientific community dealing with geohazards. To support this statement, we had already inserted the reference to the work of De Natale et al. (2006). To meet the Reviewer's request, we have provided further information highlighting the importance of the region at global scale, together with a new reference to the valuable and recent study of Troise et al. (2019) focusing on this topic.

**Line 52-53:** And then what? What are you trying to say? What can we learn from these deaths?

**Author response:** This sentence has been eliminated because considered redundant with respect to the previous text. Besides this, such deaths highlight that landslides in the area can lead to serious human consequences posing a relevant societal risk, as also remarked in previous studies (e.g., Calcaterra et al., 2003a,b; Cascini et al., 2008, cited in the manuscript).

**Figure 1:** Its global location is unknown. It is recommended to add latitude and longitude to add elevation data, now do not know the region's topography, topographic conditions, and the water system's distribution.

**Author response:** The Figure 1 has been modified in accordance with the Reviewer's suggestions.

**Data and methods**

**Line 111:** Why do you choose this time period? What about other times?

**Author response:** This time period depends on the data availability. We have not found more recent datasets.

**Line 127:** Why is there no comparison with other categories? Would it be accidental to use this category? Then how do we control the quality of data?

**Author response:** The use of this category was not accidental. We decided to refer to the most used landslide classification among the used data sources, avoiding to modify original information with other classification schemes. About the data quality control, please see our reply to the previous comment related to the line 14. In addition, we would like to remark that we have added the new section 4.5 describing the main limitations of the CAFLAG geodatabase.

**Line 138:** I don't think this is very good statistical software. Why not use R, Python, and Matlab for analysis? How is your significance tested in EXCEL? This leads me to question the availability of data quality.

**Author response:** Statistical analyses that we have performed in this study are very basic (i.e. descriptive), as requested by ESSD. Therefore, we have decided to use a simple software like Excel that, however, allows powerful data visualization and analysis tools for uses like the current one. The cited software packages are undoubtedly better than Excel, but we do not believe that the use of a specific software can be a valid reason to question the data quality.

**Table 1:** It is suggested that the author upload the disaster on that day and year in the form of documents to facilitate readers' visualization.

**Author response:** We have not understood this comment and, specifically, to what document the Reviewer refers.

**Study area**

**Reviewer comment:** I miss a description of the area's water system, vegetation, soil, and population distribution, as described in the abstract.

**Author response:** The suggested information has been added within the section 3 describing the study area.

**Line 175: Lack of references.**

**Author response:** The reference had already been indicated in the next sentence (i.e., Ducci and Tranfaglia, 2005)

**Results**

**Figure 3:** Can we get more information by discussing the seasons?

**Author response:** We thank the Reviewer for this question. Seasonal information is available for a limited number of events (482 out of 2302 events) and, in our knowledge, this is not sufficient to analyse statistical relationships between the movement types and seasonality.

**Figure 4:** I don't see any buildings. Instead, I think it was taken far away from where people live.

**Author response:** In the study area, many buildings are located close to the cliff tops (in the inland part) that in these pictures are not visible. We have modified the related sentence to clarify this.

**Figure 5:** It is recommended to take the photo from the same angle, choosing the same reference object for easy comparison.

**Author response:** We thank the Reviewer for this suggestion. In this case, photos refer to different sites (see Figure 1) and, for this reason, a common reference object could had not been identified.

**Line 210-213:** What do we learn from these statistics?

**Author response:** These statistics can help the reader in understanding the relationships between the spatial distribution of the catalogued landslides and lithological properties of the study area. The main finding highlighted by these statistics is that failures have affected mostly the rocky cliffs. This information can help local decision-makers to plan effective strategies for hazard and risk mitigation.

**Line 249:** From what? On what basis?

**Author response:** In order to support this statement, two suitable references have been added.

**Section 4.3:** So, what useful information can we draw from this? In addition, the content order of the result part needs to be modified, and there is no logic among the four results.

**Author response:** We thank the Reviewer for this suggestion. We have modified the order of contents included within the results section, in a way to present information on landslides and their impact first, and data related to the affected sites later. The new section 4.2 summarizes temporal information available for 517 out of the 2302 inventoried events. As highlighted also by Reviewer #3, this is a weak point of the CAFLAG geodatabase depending on the used data sources. In fact, most of them consisted of geomorphological inventories developed mostly by means of photo interpretation and field work (e.g., IFFI, inventory map of the AdB, CARG project) which do not allow to determine accurate temporal information on the occurrence of landslides. This limitation has been described in the new section 4.5. In the "Temporal distribution" section, instead, we have outlined the available information which can be also useful to evidence this bias.

**Figure 10:** I suggest adding a trend line for intuitive analysis.

**Author response:** As highlighted also by the Reviewer #3, this plot is affected by a bias towards more recent landslides depending on the increasing availability of documental sources over time, rather than a real rise in mass movements. This has been pointe out in the new section 4.5. Therefore, we suggest to do not add further information, such as a trend line.

**Line 277:** What do we get from the F-N curve?

**Author response:** The F-N curves show the annual frequency F of events causing N or more fatalities against the number N of fatalities. The F-N plots can be considered as representative of the current societal risk (Christian, 2004), whose concept is based on society's aversion to high-fatality incidents. The curve obtained for the study area can be of particular support for public authorities in charge of landslide risk mitigation by

emergency plans and warning systems, also in order to define acceptable and tolerable landslide risk thresholds by society.

**Line 279:** Why choose this time period? It's not in the scope of your dataset.

**Author response:** The time span 1640-2006 is referred to the historical landslide database and F-N curve developed by Cascini et al. (2008) that has been cited for comparison.

**Summary and conclusions**

**Line 293:** This should be in the introduction.

**Author response:** We have preferred to delete this sentence because it has been considered not useful within the introduction section.

**Line 312:** How do you reach this conclusion? It is suggested to add the circulation system and the distribution map of sea level pressure in the article.

**Author response:** This conclusion is inferred from the findings published by Saviano et al. (2019) and Fortelli et al. (2021) cited in the manuscript, which are based on valuable data collected by weather stations and buoys of local monitoring networks. Given that the reader can find all the related information within these articles, we suggest to do not include data on the circulation system and maps of sea level pressure here.

**Line 317:** How is this conclusion known? Urban maps of people killed by landslides should be added.

**Author response:** This conclusion has been derived from the analysis of the used data sources listed in the section 2.1. We believe that urban maps of people killed by landslides represent a degree of detail that is out of the aim of this article, since this is not focused on the analysis of the human consequences of landslides. Fatalities data that we have provided in the geodatabase can be useful to perform future analyses.

**essd-2022-267, by Esposito and Matano**
**Author Response 1st revision**

**Point-by-point response to the Reviewer #2**

**Reviewer comment:** I think it is a very good study.

**Author response:** We are grateful to the Reviewer #2 for his/her appreciation.

**Point-by-point response to the Reviewer #3**

**General comments**

**Reviewer comment:** The paper presents an Inventory of historic landslides from a region in Italy based on a compilation of various datasets such as previously published landslide inventories and historic sources.
The paper is well written and the landslide inventory is undoubtfully relevant for local planning and mitigation purposes. However, i'm not sure if the area/scope of the database is of relevance to a general geoscience community as is the scope of ESSD "high-quality data of benefit to Earth system sciences". Either the authors needs to thourroughly make the relevance much clearer or the paper is not fit for publishing in ESSD. Furthermore the database needs more explanation and documentation/discussion as mentioned below and in the specific comments which means it also has short commings both within "well-documented and highly useful data products "

**Author response:** We are grateful to the Reviewer #3 for his/her careful review of our manuscript and for the valuable comments. In accordance with the provided suggestions, in the "summary and conclusions" section we have remarked the relevance of the CAFALAG geodatabase for both local and international geoscience community. In particular, as also explained in the following comments, this geodatabase may have a significant importance to enhance the scientific knowledge on the role played by landslides in the geomorphic evolution of volcanic landscapes, and on processes that lead to slope failures and control their spatial variations. For example, data reported within this geodatabase may be useful for: i) geomorphologists and engineering geologists focusing their research on numerical models aimed to understand relationships between landslides and topographic, geotechnical and other environmental factors controlling the stability of hillslopes, in this case made by volcanic rocks, for prediction purposes; ii) scientists interested in evaluating the role played by marine factors (e.g., sea wave action) in triggering failures of coastal slopes, that in volcanic islands represent a major hazard also because of the possible cascading tsunami processes. In order to inform the reader about potentialities and limitations of the CAFLAG geodatabase, we have added a new section explaining this. Information included in this section will be essential to make the reader aware of the possible uses of this dataset. In addition, we would also remark that as pointed out by Guzzetti et al. (2012), it has been estimated that landslide inventory mapping in terrestrial settings covers only around 1% of the land surface. This situation also pushed us to publish a landslide geodatabase on a public repository, and hopefully on ESSD, with the aim of providing a scientific support to both a broad research community and local public bodies engaged in the landslide risk assessment and mitigation.

**Reviewer comment:** We need a discussion of biases in the dataset: temporal (old slides might be under reported??) volumetric: (large landslides might be overrepresented) etc.

**Author response:** In accordance with the Reviewer's suggestion, we have added the new section 4.5 "Limitations of the CAFLAG geodatabase" providing a discussion of biases in the dataset.

**Reviewer comment:** How complete is the landslide? what is your estimate, can you quantify

**Author response:** Please, see the previous comment. In the new section 4.5 we have provided some information about the completeness of the geodatabase, and a comparison with other historical databases available in the world. We have remarked that the CAFLAG geodatabase is characterized by a relatively high

spatial completeness, compared to other databases covering comparable time ranges and spatial extents even larger. It has not been possible to provide an objective quantification of the completeness due to the lack of another independent and confirmed complete inventory, or a theoretical landslide probability density function for statistical comparison.

**Reviewer comment:** Compare dataset to other datasets/landslides published in ESSD and elsewhere.

**Author response:** Please, see the previous comment. In addition, we would like to remark that in ESSD we have not found historical landslide databases based on literature and archive sources comparable with the CAFLAG one.

**Reviewer comment:** you mention climate ongoing climate change in the abstract but not in the text. You need to explain why/how climate change will affect landslides in your area and how your database can be applied to mitigate this (if possible). Alternatively remove the mention of climate change. ut i think that would be a shame not to include.

**Author response:** We thank the Reviewer for this comment and for inviting us to explain this point in the article. However, we have decided to do not address the effects of the climate change on landslides hear, but within a further article that we would like to publish throughout the next year. Therefore, we propose to remove the mention of climate change from the abstract.

**Reviewer comment:** You need to explain/justify (much more) why this dataset is relevant to a wider geoscience community (outside a purely local audience).

**Author response:** As requested, in the discussion section we have added more text to explain how this dataset could be exploited by both local and broad geoscience community. Specifically, we have pointed out that the CAFLAG geodatabase could be used to: i) enhance scientific knowledge about the role played by landslides in the geomorphic evolution of volcanic landscapes, and on processes that lead to slope failures and control their spatial distribution; ii) calibrate and validate statistically-based landslide susceptibility models; iii) assess risk conditions and support the planning of mitigation measures.

**Specific comments**

**Abstract**

**Line 6:** Very important point in deed.

**Author response:** Thanks.

**Line 13:** formation?

**Author response:** We prefer to use "modification".

**Lines 20-21:** Is this real or a bias of the data? discuss in the paper.

**Author response:** This is a bias of the data. Accordingly, we have corrected both this sentence and the related paragraph into the "Temporal distribution" section. Please, see also the responses to the previous comments related to the bias and completeness of the data.

**Lines 22-23:** As one would expect the record of landslide fatalities to be more complete that the record of all landslides maybe this demonstrates that there is a bias in the dataset skewing the data towards more recent landslides? see my previous comment.

**Author response:** Please, see the responses to the previous and following comments about the bias in the dataset. This issue has been discussed in the new section 4.5 "Limitations of the CAFLAG geodatabase".

**Line 25:** how do you propose this is done?

**Author response:** Please, see the previous comment.

**Line 29:** maybe "especially" in steadt?

**Author response:** The suggested correction has been applied.

**Lines 33-38:** You don't mention, report or discuss lahars in this paper so i dont think it is relevant to include this here.

**Author response:** In this section, we are presenting all the possible mass wasting processes occurring in volcanic settings. Even though we do not report lahars in the presented geodatabase, we believe that it is important to mention these processes since they are expected both in the Campi Flegrei and in other parts of the region in case of eruption. For this reason, we propose to do not remove this part.

**Line 41:** population number or density?

**Author response:** We have substituted "highly" with the more suitable "densely" term.

**Line 46:** I guess since pre historical times?

**Author response:** The sentence has been rephrased because of mistakes in the used terms.

**Line 48:** place name on fig 1

**Author response:** This locality is not included within the Figure 1.

**Line 49:** place name on fig 1

**Author response:** The suggested correction has been applied.

**Line 50:** place name on fig 1

**Author response:** The suggested correction has been applied.

**Line 68:** "a GIS"

**Author response:** The suggested correction has been applied.

**Figure 1:** Make the map bigger and the italy inset smaller. Harmonize with place names in text. Show absolute elevation (100 m contours or so) Maybe show simplified geology as background to support the setting paragraph. Show also an outline of what the study area is (Campi Flegrei) i guess it is not defined by the map extent?

**Author response:** In accordance with the Reviewer's suggestions, the Figure 1 has been modified as follows: 1) Italy has been placed in a smaller inset; 2) we have inserted the place names cited throughout the article; 3) we have added a colour scale to represent elevation data, and blue lines to highlight the drainage network; 4) UTM coordinates have been plotted on the map. It is worth noting that the map extent defines the Campi Flegrei study area.

**Line 106:** Why a point shapefile? dataset would have been much more usefull if you had digitized the area of the landslide where possible.

**Author response:** We agree with the Reviewer's comment. Landslides represented by polygons are certainly better than points. In this case, however, most of the used data sources did not provide polygons but only coordinates of the landslide-affected sites. In order to do not generate bias in the landslides location and extent (e.g., with polygon larger or smaller than the real affected area), we decided to use a point shapefile and to do not digitize uncertain areas.

**Table 1:** what about areas? Affected area (total area) Scarp area deposit area
These are very important to determine the magnitude/consequence of the event.

**Author response:** Please, see the previous comment.

**Table 1:** how estimated? what are the errors? from fig 6 it looks like the volumes are binned (fx 90 or so in the 10m3 size but this might of course be because the x axis is log.

**Author response:** In this work, we did not estimate areas and volumes of landslides. We only collected information already available from the used data sources. For this reason, we are not able to provide errors associated with the reported volumes. However, volumetric information displayed in Figure 6 is consistent with the order of magnitude of rock volumes displaced by landslides in the study area. Esposito et al. (2020), for example, analysed failures affecting a coastal cliff in a time span of three years. They estimated that 90th percentile of the volumetric distribution related to the detached rock blocks was equal to 2.35 m$^3$, and quantified in 150 m$^3$ the largest displaced block, in line with volumetric data shown in Figure 6.

**Table 1:** just out of interest: can you always assign a trigger with high confidence?

**Author response:** Triggering factors were indicated for a few records only, such as when the available information was so accurate to be reported (e.g., from scientific articles or field surveys).

**Table 1:** Could be interresting with a collumn stating the preconditioning factor ( type of weak geology etc.)

**Author response:** We thank the Reviewer for this suggestion. In the geodatabase, we have included geotechnical properties for many landslide-affected sites, as well as the degree of landslide hazard or susceptibility.

**Line 149:** show on fig 1 (is it the Gulf of Pozzuoli? use same name)

**Author response:** Thanks for this comment. We have changed "Gulf of Pozzuoli" to "Pozzuoli Bay" within the Figure 1.

**Line 150:** Show on map?

**Author response:** We have removed this sentence because irrelevant in the description of the study area.

**Line 168:**

**Author response:** The suggested correction has been applied

**Line 169:**

**Author response:** The suggested correction has been applied

**Line 181:** write out for clairity "maximum 10 minutes..."

**Author response:** The suggested correction has been applied

**Figure 2:** It is quite self evident just from the map distribution whether they are coastal or inner slope. and thus a bit irrelevant. It would be more interresting to display the type of landslide and maybe also the magnitude (vol) with the point size. This would make one start tinking and highlight the appeal of your dataset for others.

**Author response:** We agree with the Reviewer's suggestion to display the type of landslide movement in the Figure 2. Given that the volumetric information was available for only 277 out of the 2302 catalogued landslides, we prefer to do not represent this incomplete information.

**Figure 2:** show absolute elevation (maybe as a colour range behind the hillshade?)

**Author response:** We thank the Reviewer for this suggestion. We have tried to represent elevation by using different sets of colour ranges behind the hillshade, as suggested. In all cases, the coloured background has weakened visibility of points generating confusion in the map readability. For this reason, we propose to do not add coloured layers to the grayscale hillshade. Information on the absolute elevation can be achieved from the new version of Figure 1.

**Figure 3:** is the "unknown" binned with the complex or what? do you always know the kinematic of a historic slide?

**Author response:** As already explained into the text (lines 195-198), data sources provide information on the movement type for all the 2302 catalogued landslides (i.e., fall, flow, slide, complex). However, among the complex landslides (1115), information on the partial movement (e.g., debris slide + flow) is available for only 343 out of the 1115 complex landslides. In other words, data sources of 772 complex events do not report further information besides the complex typology. For this reason, we have decided to split the complex landslides in two categories: "complex" and "complex (with info)". This has allowed us to do not discard information available for the 343 complex events.

**Figure 4:** show on fig 1 or 2

**Author response:** As suggested, the affected localities have been indicated in Fig. 1

**Figure 5:** indicate position on fig 1 or 2

**Author response:** As suggested, the affected localities have been indicated in Fig. 1

**Line 220:** "of the 277 landslides"

**Author response:** The suggested correction has been applied.

**Figure 6:** so i guess this means that a lot of the landslides were estimated to be c. 10m3?

**Author response:** The graph shows that 90% of the landslides were estimated to displace rock volumes within 200 m$^3$

**Figure 6:** I think this fig would be a lot easier if the x axis was not log. I dont know if there is a reason to make i log (convention or such) but otherwise change it. Change caption to "… of the 277 landslides with volumetric data"

**Author response:** We agree with the Reviewer's comment. Accordingly, we have changed the format of the x-axis and the caption. The new graph shows, in addition, that landslides with volumes below 10 m$^3$ are strongly underreported.

**Lines 229-230:** i guess this also is to do with the aspect/area distribution in the field area? or what? is the aspect completely evenly distributed in the area? this should be mentioned discussed.

**Author response:** We agree with the Reviewer's comment. Accordingly, we have calculated the density of the landslides with respect to both aspect and slope distributions in the study area, in terms of events/km$^2$. Both the Figure and related text have been updated accordingly.

**Line 255:** does this mean that the remaining of te 2200+ landsldies mapped are much older than the 1820ies or are they all from within the period of the database?

**Author response:** This means that the remaining landslides are all from within the period of the database but without an accurate indication of the year of occurrence.

**Lines 256-257:** this is not what I see in Fig. 10A. there is definitely a bias towards more recent landslides being reported as i see it. This paragraph needs much more discussion of biases etc.
I dont doubt that 86,97 and 05 are years of extreme events, but you really need to discuss the completeness of the landslide db

**Author response:** Please, see the responses to the previous comments on the geodatabase completeness. We remark that this point has been discussed in the new section 4.5 "Limitations of the CAFLAG geodatabase".

**Line 266:** How about further back than the 1900? i would expect you could find records of fatalities in newspapers which should be available further back?

**Author response:** In this research, we have decided to do not go back further than 1900 in order to avoid collecting biased data affected by high uncertainty.

---

## Author Response (AR2)

**essd-2022-267, by Esposito and Matano**
**Author Response 2st revision**

**Point-by-point response to the Reviewer #1**

**Reviewer comment:** The authors have made thorough and useful revisions of their submitted manuscript. Most comments have been addressed properly in the author's reply and revised version of the manuscript. However, a few aspects still require improvement before the manuscript could be accepted for final publication.

**Author response:** We are grateful to the Reviewer #1 for his/her comments and suggestions which allowed us to improve the first version of the manuscript. Point-by-point responses to all the comments raised after the second-round revision are outlined below.

**Reviewer comment:** 1. The title says '...historical landslide events...' yet It seems to me that sections 4.1 and 4.3 are the ones that truly reflect the title of the MS. Sadly, the other valuable information that was presented in the earlier sections is not tied into the main logical thread of the MS, nor it is demonstrated how vital those may be to be used with historical landslide events. I mean the title of your article is historical landslides, but only 4.1 and 4.3 in the article really introduce the topic of historical landslides, not ask you to change the title!!

**Author response:** In the Campi Flegrei area, landslides are predominantly characterized by impulsive kinematic mechanisms and often they are not more recognizable along the slopes after a few years, because of erosion, vegetation recovery, or human interventions. At the same time, conditions that led the affected sites to be unstable remain through time. In the light of this, we remark that secondary information associated to the landslide events, and described in the sections 4.2 and 4.4, are essential to provide a complete knowledge about mass wasting processes in the area, as well as to understand the impact on people and property. From our side, hence, this information is not untied from historical landslides. In addition, in the revised version of the manuscript we have underlined that all the provided data coincide with inputs required by statistically-based landslide susceptibility models, namely "dependent" and "explanatory" variables respectively, clarifying thus their importance. This concept is well known to scientists that exploit landslide geodatabases for implementing numerical models aimed at hazard and susceptibility assessment. Our intention was hence to provide a dataset as much as complete, to be properly and widely used by other scientists.

**Reviewer comment:** 2. The presentation of the dataset is poor. I am missing at least one table/figure that really gives an overview on the whole dataset that is presented in the MS from the variables side. After all that is in the focus according to the title. The information provided in Sect. 2 is relevant, yet insufficient. The reader must get an overview on the dataset that is presented, before any detail on the measurements is discussed. It still doesn't seem to solve the problem.

**Author response:** In accordance with the Reviewer's suggestion, at the beginning of the section 3 we have added a new Figure (namely Figure 2) providing an overview of the information associated to each catalogued landslide event, together with a brief description into the text.

**Reviewer comment:** 3. I'm still confused. Is degree the recommended issue of data quality? How to check?

**Author response:** It is not clear to which degree the Reviewer is referring to. The data quality issue recommended by ESSD has been addressed into the sections 3.2 and 4.5. Specifically, spatial and temporal

accuracies have been specified for each event, as shown also in Table 1 (see "LOCAT_ACC" and "TEMP_ACC" entries) and extensively explained into the section 3.2. The most important limitations related to data quality are listed into the section 4.5.

**Reviewer comment:** 4. Can the data be updated?

**Author response:** The CAFLAG dataset has been already published on the 4TU.ResearchData web repository, and includes data until 2017. This data description paper is therefore aimed at describing such dataset only. However, we believe that updates will be possible in the next years with further research activities.

**Point-by-point response to the Reviewer #2**

**Reviewer comment:** The manuscript has improved since the initial submission. However even after the revisions I have a hard time seeing the dataset is valuable to a broader landslide-geoscience community and thus the dataset/paper falls out of the scope of ESSD of "high-quality data of benefit to Earth system sciences" The dataset may still be of use in a more local application aimed at planning countermeasures for the landslide risk reduction as suggested by the authors but not of general/broader interest.

**Author response:** We disagree with the statement of the Reviewer #2 "the dataset/paper falls out of the scope of ESSD of high-quality data of benefit to Earth system sciences". In the revised version of the article, we have provided more than one reason explaining how the CAFLAG geodatabase can benefits the Earth system sciences community, in special way researchers working in the field of landslide susceptibility/hazard analyses. We have also provided clear indications about limitations affecting the geodatabase, as commonly reported in this type of papers. We would also to point out that the CAFLAG geodatabase refers to landslides occurred in an active volcanic area. This represents a further key aspect which could benefit also researchers working in multi-hazard domains, or those engaged in assessing the geomorphic evolution of volcanic landscapes.

In addition, we would like to remark that in the scope of ESSD there is no indication about the spatial extension of datasets to be published. This is confirmed by other landslide-related data description papers published on ESSD in the last years, which refer to both local and national scales, such as:

Ardizzone, F., Bucci, F., Cardinali, M., Fiorucci, F., Pisano, L., Santangelo, M., and Zumpano, V.: Geomorphological landslide inventory map of the Daunia Apennines, southern Italy, Earth Syst. Sci. Data, 15, 753–767, https://doi.org/10.5194/essd-15-753-2023, 2023.

Luetzenburg, G., Svennevig, K., Bjørk, A. A., Keiding, M., and Kroon, A.: A national landslide inventory for Denmark, Earth Syst. Sci. Data, 14, 3157–3165, https://doi.org/10.5194/essd-14-3157-2022, 2022.

Hao, L., Rajaneesh A., van Westen, C., Sajinkumar K. S., Martha, T. R., Jaiswal, P., and McAdoo, B. G.: Constructing a complete landslide inventory dataset for the 2018 monsoon disaster in Kerala, India, for land use change analysis, Earth Syst. Sci. Data, 12, 2899–2918, https://doi.org/10.5194/essd-12-2899-2020, 2020.

Fan Yang, Xiaojun Guo, Lanxin Dai, Chaoyang He, Qiang Xu, and Runqiu Huang: Two multi-temporal datasets that track the enhanced landsliding after the 2008 Wenchuan earthquake. Xuanmei Fan, Gianvito Scaringi, Guillem Domènech, Earth Syst. Sci. Data, 11, 35–55, https://doi.org/10.5194/essd-11-35-2019, 2019.

**Point-by-point response to the Reviewer #3**

**Reviewer comment:** Line 19. was found. Word 'results' require the description of cause, not of the location.

**Author response:** We thank the Reviewer for this comment. We prefer to maintain the present tense, by substituting 'results' with the term 'concentrates'.

**Reviewer comment:** Line 20. ... from which most of the low-pressure systems come from. I expect you mean cyclones. but it is nor very clear.

**Author response:** In order to clarify this, we have substituted 'low-pressure' with 'storm'.

**Reviewer comment:** Line 180. 'This is visited' Better: Besides, it is visited

**Author response:** The suggested correction has been applied.

**Reviewer comment:** Line 221. 'involved in' or 'affected by'

**Author response:** In this case, 'affected by' is more suitable.

**Reviewer comment:** Line 305. lithoid rocks. I think better: solid rocks

**Author response:** The suggested correction has been applied here and at line 445.

**Reviewer comment:** Line 386. ot debris avalanches.

**Author response:** The suggested correction has been applied.

**Point-by-point response to the Reviewer #4**

**Reviewer comment:** The paper entitled "A geodatabase of historical landslide events occurred in the highly urbanized volcanic area of Campi Flegrei, Italy" shows the results of a study aimed at properly collecting relevant data dealing with landslides occurred in an area of southern Italy where different natural hazards coexist. These data, freely available online, can be profitably used by other researchers working in the field of landslide susceptibility/hazard analyses. In my opinion, the paper may become acceptable for publication with minor revisions.

**Author response:** We are thankful to the Reviewer for his appreciation of our work, and for his constructive and fruitful comments that helped us to improve the article. Point-by-point responses to all the minor comments raised after the second-round revision are outlined below.

**Reviewer comment:** First, I personally believe that the structure of the article would improve if the presentation of the "Study area" preceded the description of "Data and methods". Therefore, I suggest moving the current Section 3 immediately after the Introduction (of course, the current Section 2 would take the place of Section 3 just before the presentation of results).

**Author response:** Thanks for this comment. The suggested correction has been applied.

**Reviewer comment:** In Table 1 I'm not totally convinced about the goodness of the range (7-150), expressed in meters, concerning the attribute named AVG_THICK (i.e., the average thickness of the pyroclastic cover). Please check.

**Author response:** Thanks for this comment. The correct unit is centimetres and not meters. This has been corrected in Table 1 accordingly. We would like to remark that thickness data have been collected by the local basin authority and not by us in the current study.

**Reviewer comment:** In Figure 9, the F-N curve concerning the Campi Flegrei study area shows some mistakes. Indeed, F-N points associated with N values equalling or exceeding 5 are aligned along a horizontal line. This cannot happen considering that F is defined as the annual frequency of events causing N or more fatalities. In other words, as N values increases F values should decrease. Please amend accordingly.

**Author response:** We agree with this comment. A mistake has been committed during the F-N curve calculation. The mistake has been identified and corrected, as shown in the new Figure 10.

**Reviewer comment:** Finally, in Section 6 (line 371) I would avoid the use of the term "vulnerability" if applied to "the rocky coastline" (where a danger might locate) because this term applies only to elements at risk.

**Author response:** In accordance with the Reviewer's comment, the term "vulnerability" has been substituted with "susceptibility".